Open Access EGU

# Attribution of the accelerating increase in atmospheric methane during 2010–2018 by inverse analysis of GOSAT observations

Yuzhong Zhang[1,2,3], Daniel J. Jacob[3], Xiao Lu[3], Joannes D. Maasakkers[4], Tia R. Scarpelli[3], Jian-Xiong Sheng[5], Lu Shen[3], Zhen Qu[3], Melissa P. Sulprizio[3], Jinfeng Chang[6], Anthony A. Bloom[7], Shuang Ma[7], John Worden[7], Robert J. Parker[8,9], Hartmut Boesch[8,9]

[1]School of Engineering, Westlake University, Hangzhou, Zhejiang, China
[2]Institute of Advanced Technology, Westlake Institute for Advanced Study, Hangzhou, Zhejiang, China
[3]School of Engineering and Applied Science, Harvard University, MA, USA
[4]SRON Netherlands Institute for Space Research, Utrecht, the Netherlands.
[5]Center for Global Change Science, Massachusetts Institute of Technology, Cambridge, MA, USA
[6]Zhejiang University, Hangzhou, Zhejiang, China
[7]Jet Propulsion Laboratory, California Institute of Technology, Pasadena, CA, USA
[8]National Centre for Earth Observation, University of Leicester, UK
[9]Earth Observation Science, School of Physics and Astronomy, University of Leicester, UK

*Correspondence to*: Yuzhong Zhang (zhangyuzhong@westlake.edu.cn)

**Abstract.** We conduct a global inverse analysis of 2010–2018 GOSAT satellite observations to better understand the factors controlling atmospheric methane and its accelerating increase over the 2010–2018 period. The inversion optimizes 2010–2018 anthropogenic methane emissions and their trends on a 4º×5º grid, monthly regional wetland emissions, and annual hemispheric concentrations of tropospheric OH (the main sink of methane) also for individual years. We use an analytical solution to the Bayesian optimization problem that provides closed-form estimates of error covariances and information content for the solution. Our inversion successfully reduces the errors against the independent methane observations from the TCCON network and reproduces the interannual variability of the methane growth rate inferred from NOAA background sites. We find that prior estimates of fuel-related emissions reported by individual countries to the United Nations are too high for China (coal) and Russia (oil/gas), and too low for Venezuela (oil/gas) and the U.S. (oil/gas). We show that the 2010–2018 increase in global methane emissions is mainly driven by tropical wetlands (Amazon and tropical Africa), boreal wetlands (Eurasia), and tropical livestock (South Asia, Africa, Brazil), with no significant trend in oil/gas emissions. While the rise in tropical livestock emissions is consistent with bottom-up estimates of rapidly growing cattle populations, the rise in wetland emissions needs to be better understood. The sustained acceleration of growth rates in 2016–2018 relative to 2010–2013 is mostly from wetlands, while the peak methane growth rates in 2014–2015 are also contributed by low OH concentrations (2014) and high fire emissions (2015). Our best estimate is that OH did not contribute significantly to the 2010–2018 methane trend other than the 2014 spike, though error correlation with global anthropogenic emissions limits confidence in this result.

## 1 Introduction

Methane is the second most important anthropogenic greenhouse gas after $CO_2$, with an emission-based radiative forcing of 0.97 W $m^{-2}$ since pre-industrial times (Myhre et al., 2013). Methane is emitted to the atmosphere from a range of anthropogenic activities including fuel exploitation, agriculture, waste and wastewater treatment, and biomass burning. The main natural source is from wetlands, with minor contributions from geological seeps, forest fires, and termites. Atmospheric methane has a lifetime of 11.2 ±1.3 years against tropospheric oxidation by the hydroxyl radical (OH) (Prather et al., 2012). Minor sinks include stratospheric loss, oxidation by Cl atoms, and absorption by soils.

Unlike the steady rise in atmospheric $CO_2$, the rise of methane has taken place in fits and starts. Observations from the NOAA network (Dlugokencky, 2020) (https://www.esrl.noaa.gov/gmd/ccgg/trends_ch4/, last access: 22 June 2020) show a period of stabilization in the early 2000s, followed by a renewed growth after 2007 that has accelerated since 2014. Annual growth rates averaged 0.50% $a^{-1}$ for 2014–2018, compared to 0.32% $a^{-1}$ for 2007–2013. The growth of atmospheric methane concentrations, if continued at current rates in coming decades, may significantly negate the climate benefit of $CO_2$ emission reduction (Nisbet et al., 2019).

However, our understanding of the drivers behind the methane growth rate is still limited, preventing reliable projections for future changes. Explanations have differed for the renewed growth of atmospheric methane since 2007. A concurrent increase in atmospheric ethane has been interpreted as evidence of an increase in oil and gas emissions (Hausmann et al., 2016;Franco et al., 2016). However, the assumption that the ethane/methane emission ratio should be stable is questionable (Lan et al., 2019). Meanwhile, a concurrent shift towards isotopically lighter methane has been attributed to an increase in microbial sources either from livestock or wetlands (Schaefer et al., 2016;Nisbet et al., 2016). Worden et al. (2017) pointed out that the trend towards isotopically lighter methane could be explained by decreases in fire emissions that are isotopically heavy. Based on methyl chloroform observations, Turner et al. (2017) and Rigby et al. (2017) suggested that a decrease in the OH sink may be the cause of the methane regrowth.

To better interpret the methane budget and its recent trends, we present here an inverse analysis of global 2010–2018 methane observations from the GOSAT satellite instrument. GOSAT provides a long record (starting in 2009) of global high-quality observations of column methane mixing ratios (Kuze et al., 2016;Buchwitz et al., 2015). A number of inverse analyses previously used GOSAT observations to constrain methane emission estimates (Fraser et al., 2013;Monteil et al., 2013;Cressot et al., 2014;Alexe et al., 2015;Turner et al., 2015;Pandey et al., 2016;Pandey et al., 2017a;Miller et al., 2019;F. Wang et al., 2019a;Lunt et al., 2019;Maasakkers et al., 2019;Janardanan et al., 2020;Tunnicliffe et al., 2020;Yin et al., 2020). Maasakkers et al. (2019) used 2010–2015 GOSAT observations to optimize gridded methane emissions, global OH concentrations, and their 2010–2015 trends. They concluded that increasing methane emissions were driven mainly by India, China, and tropical

wetlands. Our analysis is based on that of Maasakkers et al. (2019) but extends it to 2018 in order to interpret the post-2014 acceleration. It implements for that purpose a number of major improvements to the Maasakkers et al. (2019) methodology including in particular (1) separate optimization of subcontinental wetland emissions to resolve their seasonal and interannual variability; (2) correction of stratospheric methane forward model biases based on ACE-FTS solar occultation satellite data (Waymark et al., 2014); (3) prior estimates of global fuel exploitation emissions using national reports submitted to the United Nations Framework Convention on Climate Change (UNFCCC) (Scarpelli et al., 2020), and (4) optimization of annual hemispheric OH concentrations.

## 2 Methods

We perform a global inversion to optimize the sources and sinks of atmospheric methane, and their 2010–2018 trends, by drawing information from GOSAT data and prior knowledge following the Bayes' rule.

We assemble the 2010–2018 GOSAT methane column observations in an observation vector $\boldsymbol{y}$ (Section 2.1), and optimize a state vector $\boldsymbol{x}$ including methane sources and sinks and their trends (Section 2.2). Prior estimates $\boldsymbol{x}_a$, which regularize the Bayesian solution, are compiled from bottom-up estimates for specific methane sources and sinks (Section 2.3). We use the GEOS-Chem chemical transport model (CTM) version 11.02 as the forward model to relate atmospheric methane to its sources and sinks (Section 2.4), and correct model biases in the stratosphere using independent satellite observations from the ACE-FTS instrument (Section 2.5). We solve the Bayesian optimization problem analytically to obtain both the posterior solution $\hat{\boldsymbol{x}}$ and its error covariance matrix $\hat{\mathbf{S}}$ , thus achieving a closed-form quantification of information content as part of the solution (Section 2.6). The inversion is evaluated by measuring its fit to observations (Section 2.7), lending confidence in the results before we analyze them in Section 3.

### 2.1 GOSAT observations

The observation vector for the inversion ($\boldsymbol{y}$) consists of column averaged dry-air methane mole fractions during 2010–2018 observed by the TANSO-FTS instrument on board the Greenhouse Gases Observing Satellite (GOSAT) (Kuze et al., 2009). The satellite is in polar sun-synchronous low-Earth orbit and observes methane by nadir solar backscatter in the 1.65 µm shortwave infrared absorption band. Observations are made at around 13:00 local solar time. We use the University of Leicester version 9 $CO_2$ proxy retrieval (Parker et al., 2020a). The retrieval has a single-observation precision of 13.7 ppb and a regional bias of 4 ppbv (Parker et al., 2020a), sufficient for a successful methane inversion (Buchwitz et al., 2015). The inversion ingests a total of 1.5 million successful GOSAT retrievals. Previous inversions often excluded high-latitude GOSAT observations because of seasonal bias, large retrieval errors at low solar elevations, and uncertainty in the role of the stratosphere (Bergamaschi et al., 2013; Turner et al., 2015;Z. Wang et al., 2017;Maasakkers et al., 2019). The exclusion of high-latitude

observations limited the capability of the inversions to resolve emissions at high latitudes such as boreal wetlands and oil/gas emissions in Russia (Maasakkers et al., 2019). Here we use an improved model bias correction scheme (Section 2.5) and include these high-latitude observations in the inversion.

**2.2 State vector**

The state vector (***x***) is the ensemble of variables that we seek to optimize in the inversion. In this work, the state vector includes (1) mean 2010–2018 methane emissions from non-wetland sources (all anthropogenic and natural emissions excluding wetlands) on a global 4º×5º grid (1009 elements); (2) linear trends of non-wetland emissions on that same grid (1009 elements); (3) monthly wetland emissions from 14 subcontinental regions (1512 elements) (Figure 1); and (4) annual-mean tropospheric OH concentrations in the northern and southern hemispheres (18 elements). The reason to treat wetland and non-wetland emissions separately is that wetland emissions have large seasonal and interannual uncertainties but relatively coherent spatial behaviors (Bloom et al., 2017). Therefore, we can use the inversion to characterize seasonal and interannual variability in wetland emissions, instead of imposing them as part of the prior estimate as was done by Maasakkers et al. (2019). The latter approach introduced substantial seasonal biases in the forward model simulation that then had to be empirically filtered out before conducting the inversion (Maasakkers et al., 2019).

Another improvement in the state vector definition relative to Maasakkers et al. (2019) is to optimize annual mean OH concentrations in each hemisphere rather than just globally. Y. Zhang et al. (2018) previously found with an observing system simulation experiment that it should be possible to constrain annual mean hemispheric OH concentrations from satellite methane observations. Patra et al. (2014) suggested that global CTMs are often biased in their interhemispheric OH gradient relative to methyl chloroform observations, and such bias would propagate to the solution for methane emissions.

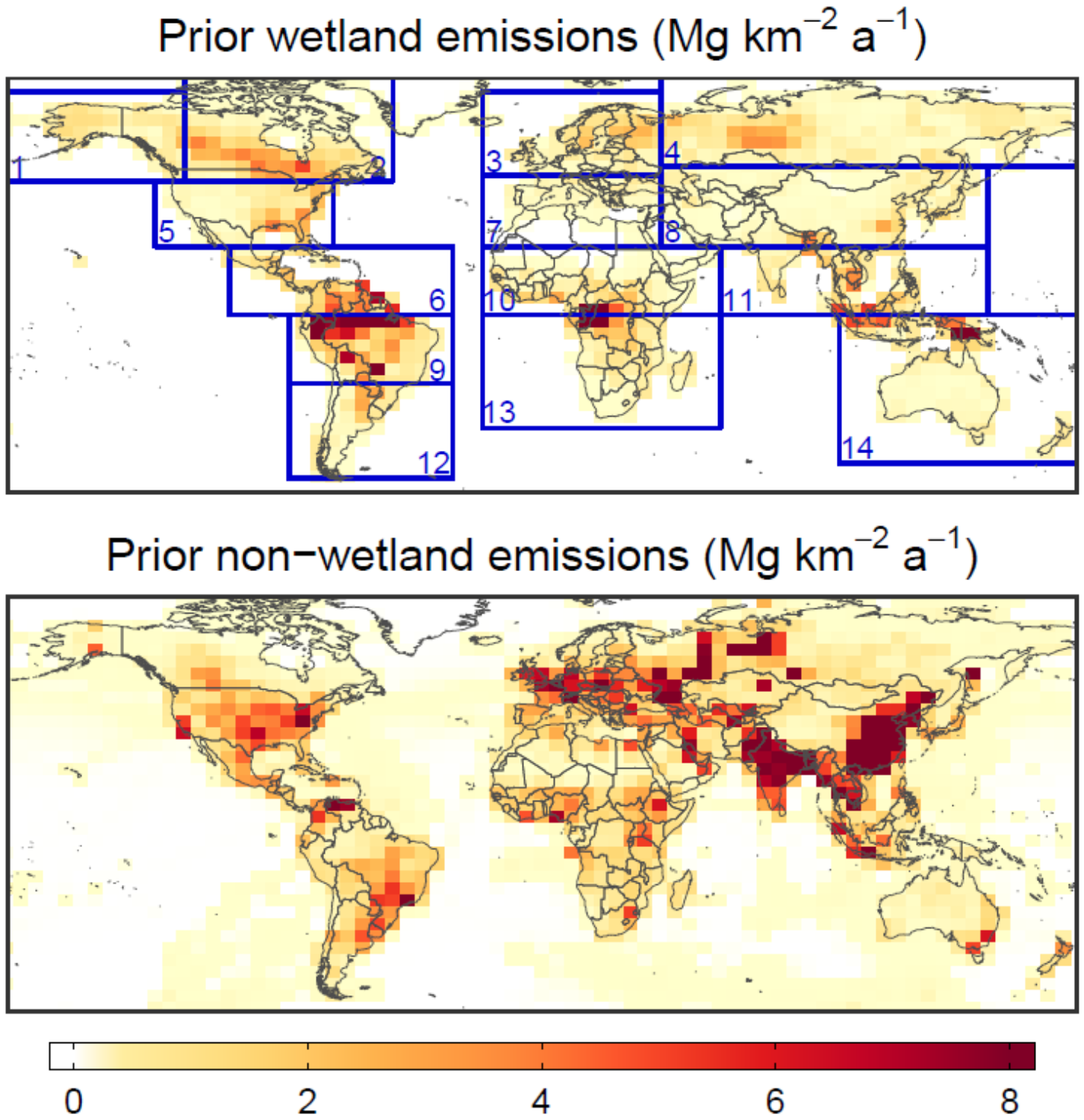

**Figure 1.** Spatial distribution of mean 2010–2018 methane emissions used as prior estimates in the inversion of GOSAT data. Blue boxes indicate the 14 subcontinental regions for which monthly wetland emissions are optimized (Section 2.2): (1) Alaska+West Canada, (2) East Canada, (3) West Europe, (4) Russia, (5) USA, (6) Latin America, (7) North Africa, (8) East Asia, (9) Amazon, (10) Sub-Sahara Africa, (11) Tropical South Asia, (12) Argentina, (13) Southern Africa, and (14) Australia.

### 2.3 Prior estimates

Prior estimates for methane sources and sinks ($\boldsymbol{x}_a$) are compiled from an ensemble of bottom-up studies. Figure 1 shows the spatial distribution of prior emission estimates. For gridded 4°×5 ° anthropogenic emissions, we use as default the EDGAR v4.3.2 global emission inventory for 2012 (https://edgar.jrc.ec.europa.eu/, last access: 1 December 2017) (Janssens-Maenhout et al., 2017). We supersede it for the US with the gridded version of the Environmental Protection Agency inventory for 2012 (Maasakkers et al., 2016). We further supersede it globally for fuel (oil, gas, and coal) exploitation with the inventory of Scarpelli et al. (2020) for 2012, which disaggregates spatially the national emissions reported to the United Nations Framework Convention on Climate Change (UNFCCC) (di.unfccc.int). All anthropogenic emissions are assumed to be aseasonal, except manure management for which we apply local temperature-dependent corrections (Maasakkers et al., 2016), and rice cultivation for which we apply gridded seasonal scaling factors from B. Zhang et al. (2016).

Monthly wetland emissions from 2010 to 2018 are from the WetCHARTS v1.0 extended ensemble mean (Bloom et al., 2017). Daily global emissions from open fires are taken from GFEDv4s (van der Werf et al., 2017), which accounts for high methane

emissions from peatland fires (Liu et al., 2020). For geological sources, we scale the spatial distribution from Etiope et al. (2019) to a global total of 2 Tg a$^{-1}$ inferred from preindustrial-era ice core $^{14}CH_4$ data (Hmiel et al., 2020). Termite emissions are from Fung et al. (1991).

The prior estimates for the hemispheric tropospheric OH concentrations are based on a GEOS-Chem full chemistry simulation (Wecht et al., 2014). The monthly 3-D OH concentration fields from this full chemistry simulation are also used in the forward model. We optimize hemispheric OH concentrations as the methane loss frequency [s$^{-1}$] due to oxidation by tropospheric OH ($k^i$ ) in the northern and southern hemispheres ($i$ = north or south):

$$k^i = \frac{\int_{\mathrm{troposphere},i} k'(T)[\mathrm{OH}] n_{\mathrm{CH_4}} \mathrm{d}v}{\int_{\mathrm{atmosphere}} n_{\mathrm{CH_4}} \mathrm{d}v} \tag{1}$$

where $n_{\mathrm{CH_4}}$ is methane number density [molecules cm$^{-3}$] , $v$ is volume, and $k'(T)$=2.45×10$^{-12}$ $e^{-1775/T}$ cm$^3$ molec$^{-1}$ s$^{-1}$ is the temperature-dependent oxidation rate constant (Burkholder et al., 2015). In this definition, the denominator of Eq. 1 integrates over the entire atmosphere and the numerator integrates over the hemispheric troposphere. Hence, global methane loss frequency (or inverse lifetime; $k$) due to oxidation by tropospheric OH can be expressed as the sum of hemispheric values ($k = 1/\tau = k^{\mathrm{north}} + k^{\mathrm{south}}$ where $\tau$ is the global lifetime due to oxidation by tropospheric OH). Our prior estimates from Wecht et al. (2014) are 0.050 a$^{-1}$ for $k^{\mathrm{north}}$ and 0.043 a$^{-1}$ for $k^{\mathrm{south}}$, which translates to a $\tau$ of 10.7 years and a north to south inter-hemispheric OH ratio of 1.16. In comparison, the methyl chloroform proxy infers $\tau$ of 11.2±1.3 years (Prather et al., 2012) and an inter-hemispheric ratio in the range 0.85–0.98 (Montzka et al., 2000;Prinn et al., 2001;Krol and Lelieveld, 2003;Bousquet et al., 2005;Patra et al., 2014), while the ACCMIP model ensemble yields a $\tau$ of 9.7±1.5 years and an inter-hemispheric ratio of 1.28±0.10 (Naik et al., 2013).

Our prior estimate assumes no 2010–2018 trends in non-wetland emissions on the 4°×5 ° grid except for interannual variability in fires (GFED4s). In this manner, all information on anthropogenic emission trends is from the GOSAT observations.

The Bayesian inversion requires error statistics for the prior estimates. The prior error covariance matrix ($\mathbf{S}_a$) is constructed as follows. For non-wetland emissions, we assume 50% error standard deviation for individual grid cells and 20% for each source category when aggregated globally. We specify an absolute error standard deviation of 5% a$^{-1}$ for linear trends of non-wetland emissions. For wetland emissions, we take the full error covariance structure (including spatial and temporal error covariance) derived from the WetCHARTs ensemble members for the 14 subcontinental regions (Bloom et al., 2017). For annual hemispheric OH concentrations, we assign 5% independent errors for individual years on top of a 10% systematic error for the multi-year mean.

## 2.4 Forward model

We use the GEOS-Chem CTM v11.02 as forward model for the inversion (Wecht et al., 2014;Turner et al., 2015;Maasakkers et al., 2019). The simulation is conducted at 4º×5º horizontal resolution with 47 vertical layers (~ 30 layers in the troposphere) and is driven by 2009–2018 MERRA-2 meteorological fields (Gelaro et al., 2017) from the NASA Global Modeling and Assimilation Office (GMAO). The prior simulation is from 2010 to 2018 with a one-year spin-up starting from January 2009 to establish methane gradients driven by synoptic-scale transport (Turner et al., 2015). We set the initial conditions on January 1, 2010 to be unbiased by removing the zonal mean biases relative to GOSAT observations. Thus we attribute any model departures from observations over the 2010–2018 period in the inversion to errors in sources and sinks over that period.

We use archived 3-D monthly fields of OH concentration from a GEOS-Chem full chemistry simulation (Wecht et al., 2014) to compute the removal of methane from oxidation by tropospheric OH. Other minor loss terms include stratospheric oxidation computed with archived monthly loss frequencies from the NASA Global Modeling Initiative model (Murray et al., 2012), tropospheric oxidation by Cl atoms computed with archived Cl concentration fields from X. Wang et al. (2019b), and monthly soil uptake fields from Murguia-Flores et al. (2018). The inversion does not optimize these minor sinks. The loss from oxidation by Cl is 5.5 Tg $a^{-1}$, accounting for ~ 1% of methane loss. This estimate by X. Wang et al. (2019b) is lower than the previous estimate of 9 Tg $a^{-1}$ (Sherwen et al., 2016) used by Maasakkers et al. (2019) and is consistent with a recent analysis of methane and CO isotopic signatures (Gromov et al., 2018). Use of monthly soil uptake fields from the Murguia-Flores et al. (2018) climatology of 2000-2009 data is another update relative to Maasakkers et al. (2019) and yields a global soil sink of 34 Tg $a^{-1}$.

## 2.5 Forward model bias correction

The GEOS-Chem simulated methane columns have a latitude-dependent background bias relative to the GOSAT data (Turner et al., 2015). This is thought to result from excessive meridional transport in the stratosphere, a common problem in global models (Patra et al., 2011). In particular, coarse-resolution global models have difficulty resolving polar vortex dynamics that control the distribution of stratospheric methane in the winter-spring hemisphere (Stanevich et al., 2019). GEOS-Chem model evaluation with stratospheric sub-columns derived from ground-based TCCON column measurements shows that the stratospheric bias varies seasonally (Saad et al., 2016). Previous GEOS-Chem based inversions of GOSAT data (Turner et al., 2015;Maasakkers et al., 2019) developed correction schemes by fitting differences between the prior model simulation and background GOSAT observations as a second-order polynomial function of latitude. However, these correction schemes did not consider the seasonal variation of the stratospheric biases. Moreover, they could falsely attribute high-latitude model-GOSAT differences to stratospheric model bias rather than to errors in prior emissions.

Here we improve the stratospheric bias correction by using satellite observations from ACE-FTS v3.6 (Waymark et al., 2014;Koo et al., 2017). ACE-FTS is a solar occultation instrument launched in 2003 and measures vertical profiles of stratospheric methane (Bernath et al., 2005). We compute correction factors to GEOS-Chem stratospheric methane sub-columns as a function of season and equivalent latitude, based on the ratios of stratospheric methane sub-columns between ACE-FTS and GEOS-Chem prior simulations (Figure 2). A global scaling factor (1.06) is applied to these correction factors to enforce mass conservation for methane in the stratosphere, so that the correction does not introduce a spurious stratospheric sink in the model simulation. We use equivalent latitude, computed on the 450 K isentropic surface from MERRA-2 reanalysis fields, as one of the predictors for parameterization. The equivalent latitude is a potential vorticity (PV) based coordinate that maps PV to latitude, based on areas enclosed by PV isopleths (Butchart and Remsberg, 1986), and is often used to represent the influence of high-altitude dynamics (e.g., polar vortex) on chemical tracers (e.g., Engel et al., 2006;Hegglin et al., 2006;Strahan et al., 2007). Figure 2 shows that GEOS-Chem model biases tend to be large at high latitudes of the winter-spring hemisphere, and are small in the tropics and in the summer-fall hemisphere. Based on information provided by ACE-FTS observations, the model bias correction scheme is able to capture these seasonal and latitudinal variations, which are not resolved by a second-order polynomial function used in Turner et al. (2015) and Maasakkers et al. (2019). Furthermore, we can attribute the stratospheric bias as specifically due to the polar vortex dynamical barrier being too weak in the model.

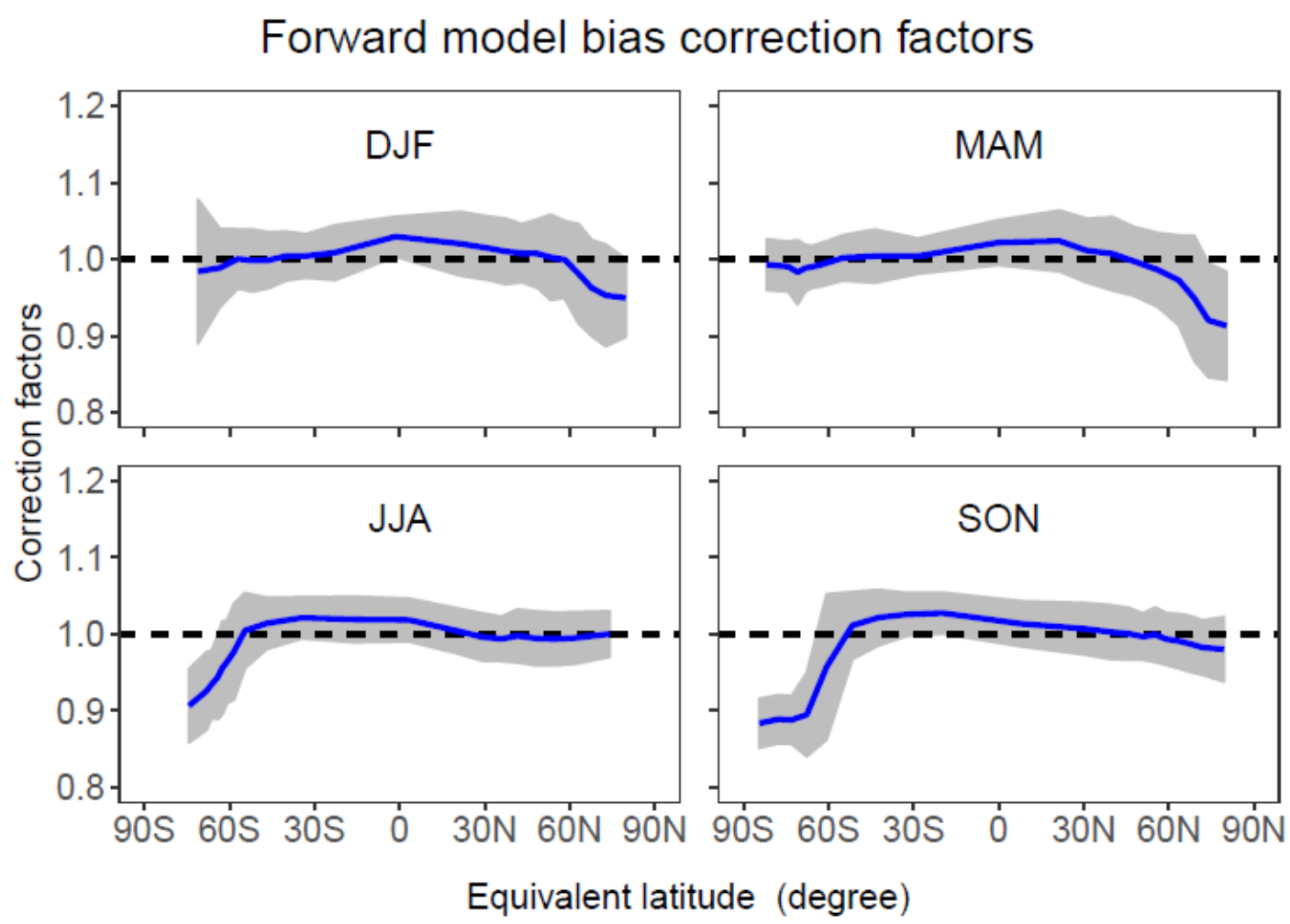

**Figure 2.** GEOS-Chem stratospheric bias correction based on ACE-FTS observations. Correction factors for stratospheric subcolumns are shown in blue lines as a function of season and equivalent latitude. Grey shading represents the fitting uceratinty. The correction factor is signficant poleward of 60 degrees in winter-spring, consistent with model error in accounting for polar vortex dynamics.

## 2.6 Inversion procedure

We perform the inversion by minimizing the Bayesian cost function (Brasseur and Jacob, 2017):

$$J(\boldsymbol{x}) = (\boldsymbol{x}-\boldsymbol{x}_{\mathrm{a}})^{\mathrm{T}}\mathbf{S}_{\mathrm{a}}^{-1}(\boldsymbol{x}-\boldsymbol{x}_{\mathrm{a}}) + \gamma(\boldsymbol{y}-\mathbf{K}\boldsymbol{x})^{\mathrm{T}}\mathbf{S}_{\mathrm{O}}^{-1}(\boldsymbol{y}-\mathbf{K}\boldsymbol{x}) \quad (2)$$

Here $\mathbf{S}_{\mathrm{O}}$ is the observation error covariance matrix including contributions from the instrument error and the forward model error. $\mathbf{S}_{\mathrm{O}}$ is taken to be diagonal and the variance terms are computed with the residual error method of Heald et al. (2004) as applied to GOSAT observations by Turner et al. (2015) and Maasakkers et al. (2019). The observational error standard deviation averages 13 ppbv. The Jacobian matrix $\mathbf{K} = \frac{\partial \boldsymbol{y}}{\partial \boldsymbol{x}}$ that relates $\boldsymbol{y}$ (observations) to $\boldsymbol{x}$ (state vector) is a linearized description of the forward model. We explicitly compute the Jacobian matrix by perturbing each individual element of $\boldsymbol{x}$ independently in GEOS-Chem and calculating the sensitivity of $\boldsymbol{y}$ to that perturbation. $\boldsymbol{x}_{\mathrm{a}}$ is the prior estimate for $\boldsymbol{x}$ and $\mathbf{S}_{\mathrm{a}}$ is the prior error covariance matrix (Section 2.3). $\gamma$ is the regularization parameter taken to be 0.05 following Y. Zhang (2018) and Maasakkers et al. (2019) to account for missing error covariance structure in $\mathbf{S}_{\mathrm{O}}$.

Minimizing $J(\boldsymbol{x})$ (Eq. 2) by solving $\mathrm{d}J/\mathrm{d}\boldsymbol{x}=0$ analytically (Rodgers, 2000; Brasseur and Jacob, 2017) yields a best posterior estimate of the state vector ($\hat{\boldsymbol{x}}$) and the associated posterior error covariance matrix ($\hat{\mathbf{S}}$) characterizing the error statistics of $\hat{\boldsymbol{x}}$:

$$\hat{\boldsymbol{x}} = \boldsymbol{x}_{\mathrm{a}} + (\gamma\mathbf{K}^{\mathrm{T}}\mathbf{S}_{\mathrm{O}}^{-1}\mathbf{K} + \mathbf{S}_{\mathrm{a}}^{-1})^{-1}\gamma\mathbf{K}^{\mathrm{T}}\mathbf{S}_{\mathrm{O}}^{-1}(\boldsymbol{y}-\mathbf{K}\boldsymbol{x}_{\mathrm{a}}), \quad (3)$$

$$\hat{\mathbf{S}} = (\gamma\mathbf{K}^{\mathrm{T}}\mathbf{S}_{\mathrm{O}}^{-1}\mathbf{K} + \mathbf{S}_{\mathrm{a}}^{-1})^{-1}, \quad (4)$$

From there we derive the averaging kernel matrix $\mathbf{A} = \partial\hat{\boldsymbol{x}}/\partial\boldsymbol{x}$ describing the sensitivity of the solution to the true state

$$\mathbf{A} = \mathbf{I} - \hat{\mathbf{S}}\mathbf{S}_{\mathrm{a}}^{-1}. \quad (5)$$

The trace of the averaging kernel matrix is referred to as the degrees of freedom for signal (DOFS) (Rodgers, 2000) and represents the number of independent pieces of information on the state vector that are constrained by the inversion.

The posterior solution is often presented in reduced dimensionality. For example, spatially resolved emission and trend estimates on the 4°×5° grid can be aggregated to countries or regions, or to global/regional emissions from individual source sectors (e.g., oil/gas, livestock, etc.). Let **W** be a matrix that represents the linear transformation from the full state vector to a reduced state vector. The posterior estimation of the reduced state vector ($\hat{\boldsymbol{x}}_{\mathrm{red}}$) is computed as

$$\hat{\boldsymbol{x}}_{\mathrm{red}} = \mathbf{W}\hat{\boldsymbol{x}}. \quad (6)$$

with posterior error covariance matrix

$$\hat{\mathbf{S}}_{\mathrm{red}} = \mathbf{W}\hat{\mathbf{S}}\mathbf{W}^{\mathrm{T}} \quad (7)$$

and averaging kernel matrix

$$\mathbf{A}_{\mathrm{red}} = \mathbf{W}\mathbf{A}\mathbf{W}^{*} \quad (8)$$

where $\mathbf{W}^{*} = \mathbf{W}^{\mathrm{T}}(\mathbf{W}\,\mathbf{W}^{\mathrm{T}})^{-1}$ is the pseudo-inverse of **W**. The advantage of this approach is that the derived regional/global budget terms and their error covariance structures are consistent with the full inversion. In the case of aggregation by sectors, we construct **W** on the basis of the relative contribution of the sector to the prior emissions in each 4°×5° grid cell. This does

not assume that the prior distribution of sectoral emissions is correct, only that the relative allocation within a given 4°×5° grid cell is correct.

### 2.7 Evaluation of posterior simulation

We conduct a posterior simulation driven by the optimized (posterior) distributions of methane emissions, emission trends, and OH concentrations to evaluate the inversion. The posterior simulation results are compared with the training data (GOSAT) as well as evaluation data including TCCON total column measurements (tccondata.org) (Wunch et al., 2011) and NOAA surface measurements (www.esrl.noaa.gov/gmd/ccgg/flask.php) (Dlugokencky et al., 2020). Figure 3 shows the GEOS-Chem comparison to the GOSAT data. As expected for a successful inversion, the posterior simulation achieves a better fit to GOSAT observations than the prior simulation, both spatially and temporally. The prior simulation has a negative bias that grows with time and is particularly large in the extratropical northern hemisphere and tropics. The prior biases have a large seasonal structure presumably associated with errors in wetland emissions. All these features largely vanish in the posterior simulation through the optimized adjustment of the state vector.

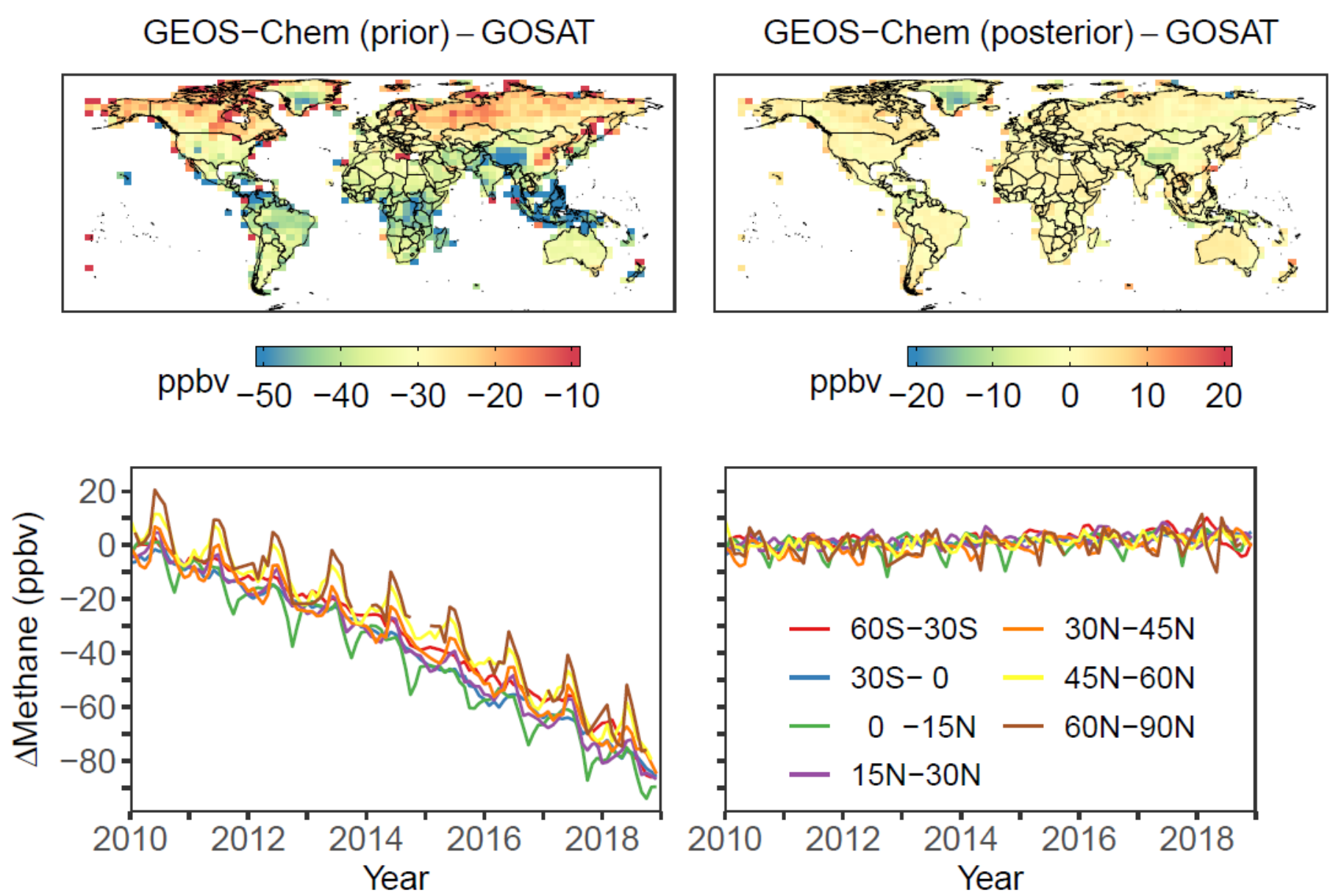

**Figure 3.** Difference of methane columns between GEOS-Chem simulations and GOSAT observations. Results are shown for GEOS-Chem using prior (left) and posterior (right) state vector estimates, and for spatial distribution averaged during 2010–2018 (top) and monthly time series of zonal means in different latitude bands (bottom). Note different color scales in the top panels. Tick marks of *x* axes in bottom panels represent January of each year.

Figure 4 presents evaluations against independent 2010–2018 observations from TCCON and NOAA sites, arranged by latitude. Values are shown as the root mean square error (RMSE) for individual sites. Figure 4 shows that the inversion substantially improves the agreement between simulations and observations for all TCCON sites and almost all NOAA surface sites. Average root-mean-square errors are reduced by 60% for TCCON sites (prior: 38 ppbv; posterior: 15 ppbv) and by 33% for NOAA surface sites (prior: 42 ppbv; posterior: 28 ppbv). The seasonal component of the errors (root-mean-square errors computed from monthly mean model-observation differences after annual mean biases are removed, not shown in figure) are reduced on average by 44% for TCCON sites (prior: 6.6 ppbv; posterior: 3.7 ppbv) and 27% for surface sites (prior: 11 ppbv; posterior: 8 ppbv), primarily owing to optimized seasonal variations in wetland emissions.

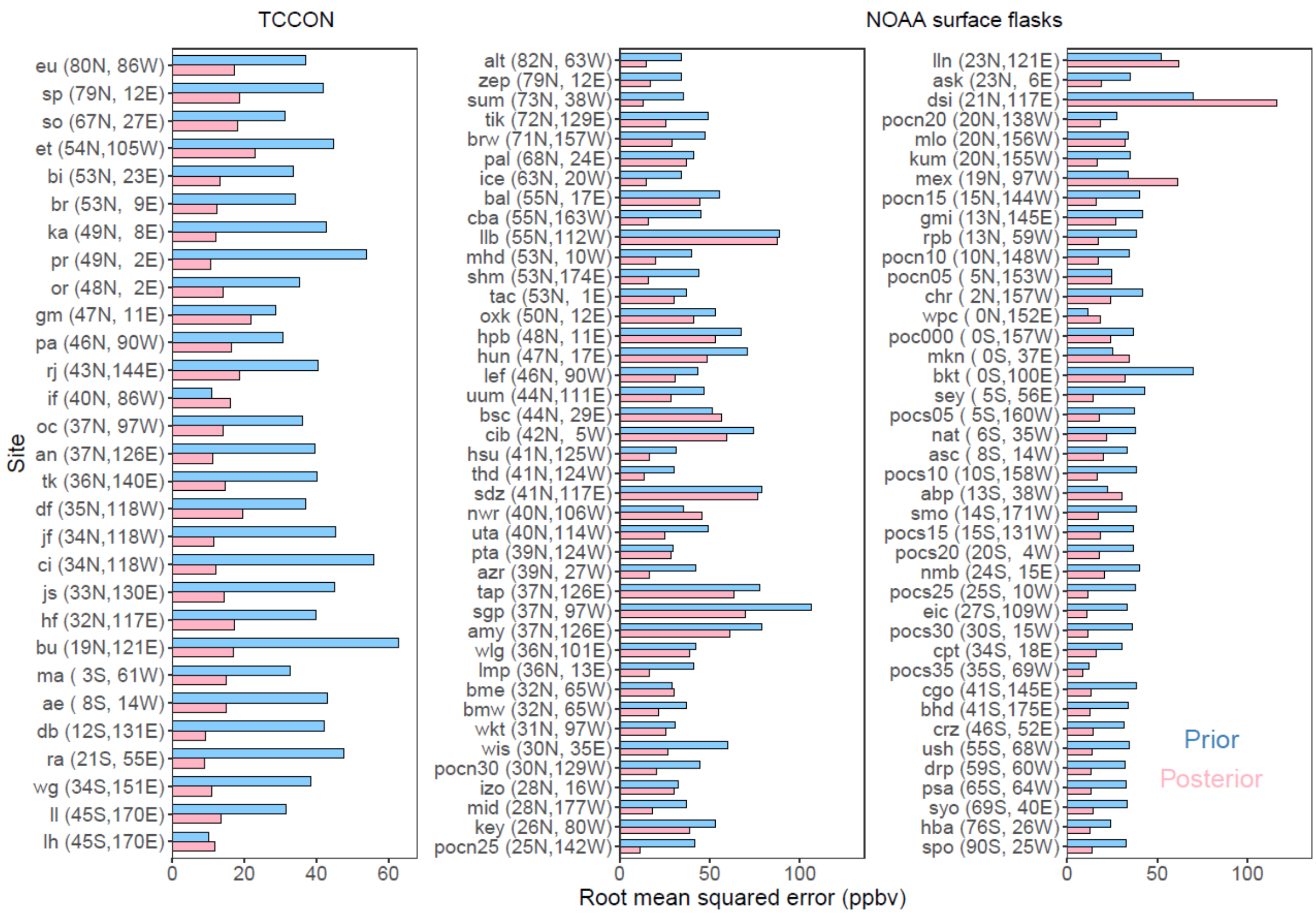

**Figure 4.** Root mean square errors of prior and posterior GEOS-Chem simulations relative to TCCON observations of dry column methane mixing ratios (left) and NOAA observations of surface air mixing ratios (middle and right). Observation sites are arranged by latitude. Data are for 2010–2018. Site names are shown along with their latitude and longitude (more information about these sites can be found at tccon-wiki.caltech.edu/Sites and www.esrl.noaa.gov/gmd/dv/site/index.php?program=ccgg). A mountain-top TCCON site located at Zugspitze, Germany (zs; ~ 3000 m.a.s.l.) is excluded because the terrain effect on the total column is not resolved by the coarse resolution model.

Atmospheric Chemistry and Physics
EGU
Discussions

## 3 Results and discussion

### 3.1 Anthropogenic emissions

Figure 5 shows the correction factors from the inversion to 2010–2018 mean non-wetland emissions (posterior/prior ratios) along with the associated averaging kernel sensitivities (corresponding diagonal terms of the averaging kernel matrix). The averaging kernel sensitivities largely follow the pattern of prior emissions and are highest in major anthropogenic source regions. We achieve 179 independent pieces of information (DOFS) for constraining the emissions on the $4^{o}\times5^{o}$ grid. By applying the posterior/prior correction factors to the prior distribution of each anthropogenic emission sector, we obtain improved estimates for anthropogenic emissions for that sector.

We find that the prior emission inventory significantly overestimates anthropogenic emissions in eastern China (Figure 5). The posterior estimate of Chinese anthropogenic emissions (48 Tg $a^{-1}$) is 30% lower than the prior estimate (67 Tg $a^{-1}$), and is also lower than the latest national report from China to the UNFCCC of 55 Tg $a^{-1}$ for 2014 (Figure 6). Based on the relative contribution of each sector in the prior inventory, we attribute ~ 60% of this downward correction to coal mining. The overestimation of anthropogenic emissions from China has been reported by previous global and regional GOSAT inversion studies using different EDGAR inventory versions as prior estimates (Monteil et al., 2013;Thompson et al., 2015;Turner et al., 2015;Maasakkers et al., 2019;Miller et al., 2019).

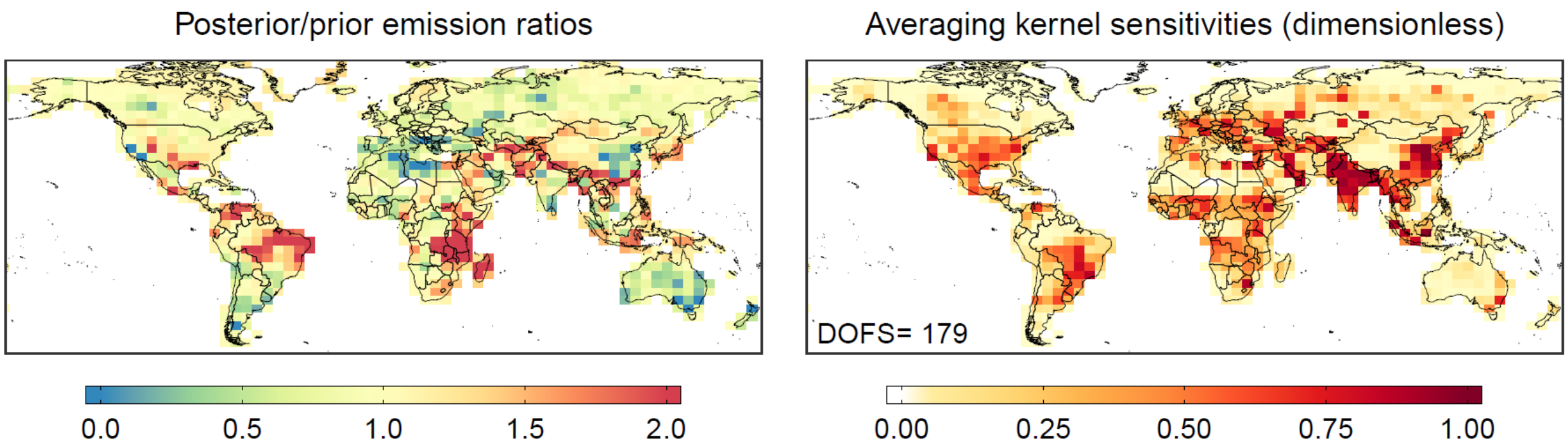

**Figure 5.** Corrections to prior estimates of 2010–2018 mean non-wetland methane emissions. (Left) posterior/prior emission ratios. (Right) averaging kernel sensitivities (diagonal elements of averaging kernel matrix). The sum of averaging kernel sensitivities defines the degrees of freedom for signal (DOFS), shown inset.

Another major downward correction is for the oil/gas producing regions in Russia. We estimate Russia's anthropogenic emissions to be 21 Tg $a^{-1}$, as opposed to the prior estimate of 36 Tg $a^{-1}$ (Figure 6), and the difference is mainly attributable to the oil/gas sector (posterior: 15 Tg $a^{-1}$; prior: 27 Tg $a^{-1}$). This finding is consistent with Maasakkers et al. (2019) though they used a different oil/gas emission inventory. Russia has been revising downwards its national emission estimates submitted to

the UNFCCC, and our posterior estimate of total anthropogenic emissions is closer to the country's latest submission to UNFCCC for 2010–2018 (16 Tg $a^{-1}$; Figure 6). The new submission revises oil/gas methane emissions downward by a factor of 3 relative to the previous submission used as prior estimate in our inversion (Scarpelli et al., 2020).

We find large upward corrections to the prior inventory over eastern Africa (Mozambique, Zambia, Tanzania, Ethiopia, Uganda, Kenya, and Madagascar) and over South America (Brazil). A previous inversion suggested that corrections for these regions may be due to an underestimation of prior wetland emissions (Maasakkers et al., 2019). Our inversion, which optimizes wetland and anthropogenic emissions separately, more specifically attributes the underestimation to livestock emissions. The upward correction pattern in Brazil is located near the country's "agricultural frontier" where forest is converted to agricultural lands for livestock and crops (Nepstad et al., 2019). Herrero et al. (2013) identified eastern Africa as the region of the highest livestock emission intensity (emission/kg protein) because of low feed efficiency, suggesting that emission factors in this region may be underestimated in bottom-up calculation given a general lack of region-specific information for developing countries.

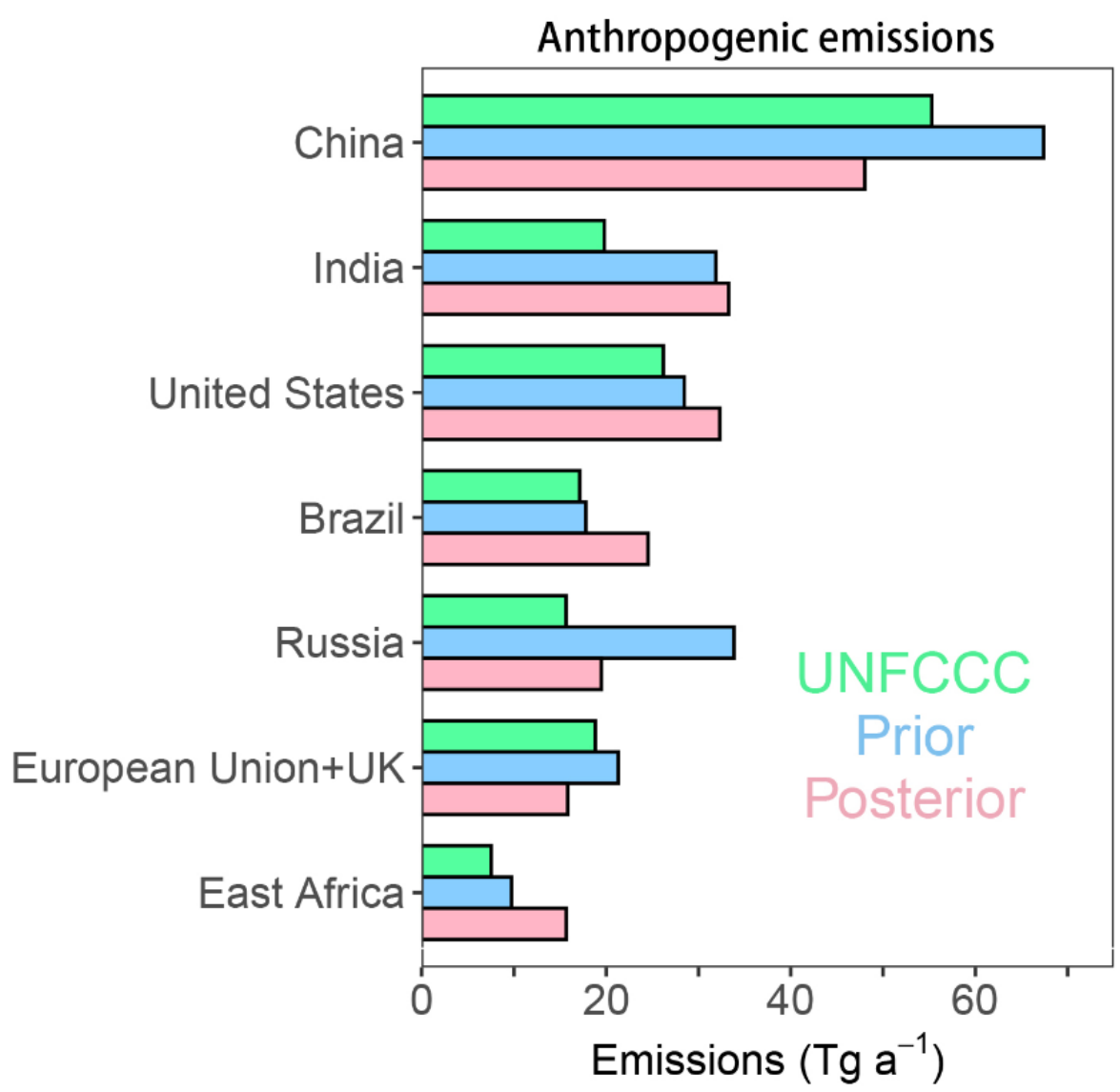

**Figure 6.** National and regional estimates of 2010–2018 mean methane emissions from anthropogenic sources. Included are the top 5 individual countries, the European Union (including the United Kingdom), and East Africa (including Mozambique, Zambia, Tanzania, Ethiopia, Uganda, Kenya, and Madagascar). The UNFCCC record is from di.unfccc.int (accessed on July 10, 2020). Non-Annex I countries do not report yearly emissions to the UNFCCC and for those we use the latest UNFCCC submissions (Brazil, 2015; China, 2014; Ethiopia, 2013; India, Madagascar, Kenya, 2010; Uganda, Zambia, 2000; Mozambique, Tanzania, 1994).

Another upward correction pattern in South America is located near Venezuela, a major oil producing country in the region. The large discrepancy in Venezuela likely reflects underestimation of fossil fuel emissions by the national estimates for 2010 reported to UNFCCC. Upward corrections are also seen in central Asia (Iran, Turkmenistan), where the regional posterior estimates (11.7±0.9 Tg $a^{-1}$) are 43% higher than the prior (8.2±1.4 Tg $a^{-1}$), with adjustments mainly attributed to the oil/gas sector. This region has previously been identified by satellite observations as a methane emission hotspot (Buchwitz et al., 2017; Varon et al., 2019; Schneising et al., 2020).

The inversion finds small upward corrections over the US (prior: 28 Tg $a^{-1}$; posterior: 32 Tg $a^{-1}$) (Figure 6), resulting mainly from underestimation of emissions from the oil/gas sector in the western and southwestern US (Figure 5). This result is broadly consistent with a number of basin-level studies based on aircraft and satellite measurements (e.g., Kort et al., 2014;Karion et al., 2015;Smith et al., 2017;Schwietzke et al., 2017;Peischl et al., 2018;Cui et al., 2019;Y. Zhang, 2020;Gorchov Negron et al., 2020; Schneising et al., 2020) and national assessments of methane emissions from oil/gas production (Alvarez et al., 2018;Maasakkers, 2020).

Small downward corrections with a diffuse pattern are found over Europe. The posterior estimate of anthropogenic emissions for the European Union (including the UK) is 16 Tg $a^{-1}$, slightly lower than the prior estimate (21 Tg $a^{-1}$) and the UNFCCC national reports (19 Tg $a^{-1}$ for 2014) (Figure 6). Source sector attribution is difficult in this case because of spatial overlap between emission sectors. The inversion finds only small adjustments to prior emissions for India (prior: 32 Tg $a^{-1}$; posterior: 33 Tg $a^{-1}$) even though the information content is relatively large, confirming the prior inventory used in the inversion. Our estimate, however, is much higher than a previous inversion study for India (Ganesan et al., 2017) (22 Tg $a^{-1}$), which found their result in agreement with India's UNFCCC report (20 Tg $a^{-1}$ for 2010) (Figure 6). The discrepancy is mainly in the livestock sector, which we find to be greatly underestimated in the UNFCCC submission. Livestock emission trends in India are discussed further below.

### 3.2 Anthropogenic emission trends

Figure 7 shows the spatial distribution of 2010–2018 trends for anthropogenic emissions inferred from GOSAT observations, along with the associated averaging kernel matrix sensitivities. The GOSAT observations provide 42 pieces of information to constrain the spatial distribution of anthropogenic emission trends. The constraints are strongest in China and India, but there is also strong information aggregated over other continental regions. The prior estimate assumed no anthropogenic trends anywhere, therefore the posterior trends are driven solely by the GOSAT information. The corresponding diagonal elements of the posterior error covariance matrix allow us to quantify error standard deviations on the results.

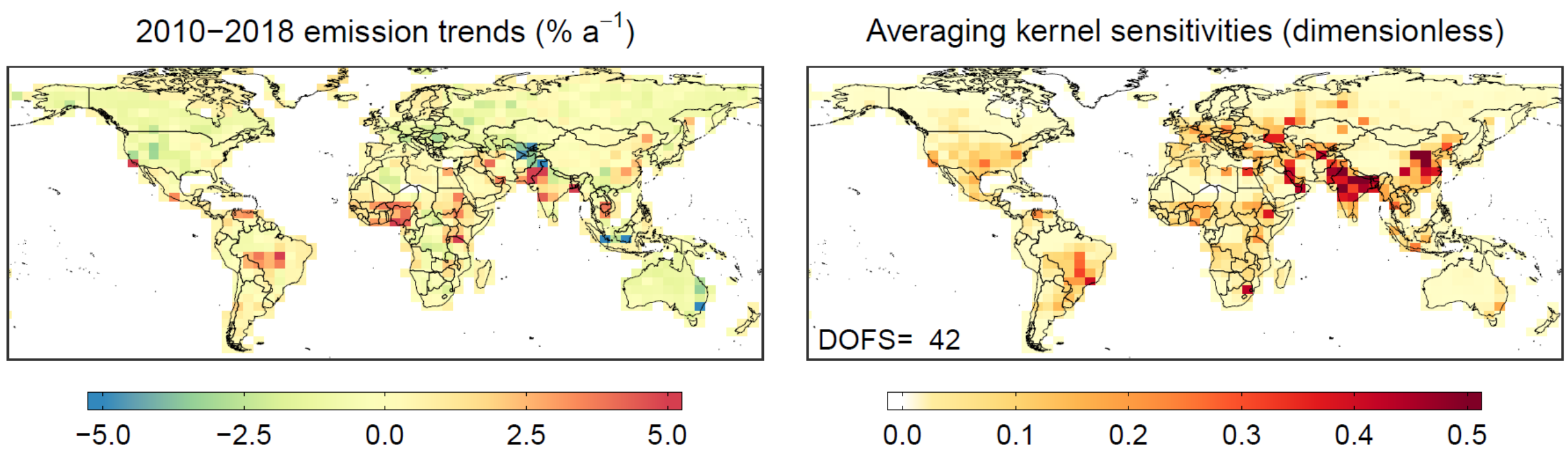

**Figure 7.** Anthropogenic methane emission trends for 2010–2018, as informed by GOSAT observations. (Left) Relative emission trends on the 4°×5° grid. (Right) Averaging kernel sensitivities. Degrees of freedom for signal (DOFS) is inset.

Significant positive trends of anthropogenic emissions are found in South Asia (0.57±0.16 Tg a$^{-1}$ a$^{-1}$ or 1.3±0.4% a$^{-1}$; Pakistan and India), East Africa (0.21±0.10 Tg a$^{-1}$ a$^{-1}$ or 1.4±0.6 % a$^{-1}$; Ethiopia, Tanzania, Uganda, Kenya, and Sudan), West Africa (0.32±0.10 Tg a$^{-1}$ a$^{-1}$ or 4.6±1.4% a$^{-1}$; Nigeria, Niger, Ghana, Cote d'Ivoire, Mali, Benin, Burkina Faso), and Brazil (0.17±0.15 Tg a$^{-1}$ a$^{-1}$ or 0.7±0.6% a$^{-1}$). All these regions are seeing rapid increases in livestock populations. According to the United Nations Food and Agriculture Office (UNFAO) statistics, the fastest growing cattle populations in the world over the 2010–2016 period were Pakistan (1.4×10$^{6}$ heads a$^{-1}$), Ethiopia (1.2×10$^{6}$ heads a$^{-1}$), Tanzania (1.1×10$^{6}$ heads a$^{-1}$), and Brazil (0.9×10$^{6}$ heads a$^{-1}$). Indeed, our sectoral attribution of the trends in Figure 7 attributes them mostly to livestock (0.40 Tg a$^{-1}$ a$^{-1}$ in South Asia, 0.13 Tg a$^{-1}$ a$^{-1}$ in East Africa, 0.15 Tg a$^{-1}$ a$^{-1}$ in West Africa, and 0.11 Tg a$^{-1}$ a$^{-1}$ in Brazil). Our inversion results for these regional trends in livestock emissions are consistent with the trends from bottom-up livestock emission inventories (FAOSTAT, IPCC tier I method; EDGAR v4.3.2 and v5, hybrid tier I method; Chang et al. (2019), IPCC tier II method), as shown in Figure 8. Anthropogenic methane emission trends in South Asia, Africa, and Brazil add up to 1.3 Tg a$^{-1}$ a$^{-1}$, which as we will show below amounts to the global anthropogenic emission trend.

A positive trend in anthropogenic emissions (0.27±0.20 Tg a$^{-1}$ a$^{-1}$ or 0.6±0.4% a$^{-1}$) is found over China driven by coal mining (northern China) and rice (southern China), but the magnitude of the trend is much smaller than previous inverse analyses of satellite and surface observations for time horizons before 2015 (Bergamaschi et al., 2013;Thompson et al., 2015;Saunois et al., 2017;Miller et al., 2019;Maasakkers et al., 2019). We infer a much stronger trend (0.72±0.39 Tg a$^{-1}$ a$^{-1}$ or 1.6±0.9% a$^{-1}$) for China if we restrict the GOSAT record to 2010–2016. Our results thus suggest that anthropogenic emission trends in China peaked midway within the 2010-2018 record. Indeed, coal production in China peaked in 2013 (Sheng et al., 2019).

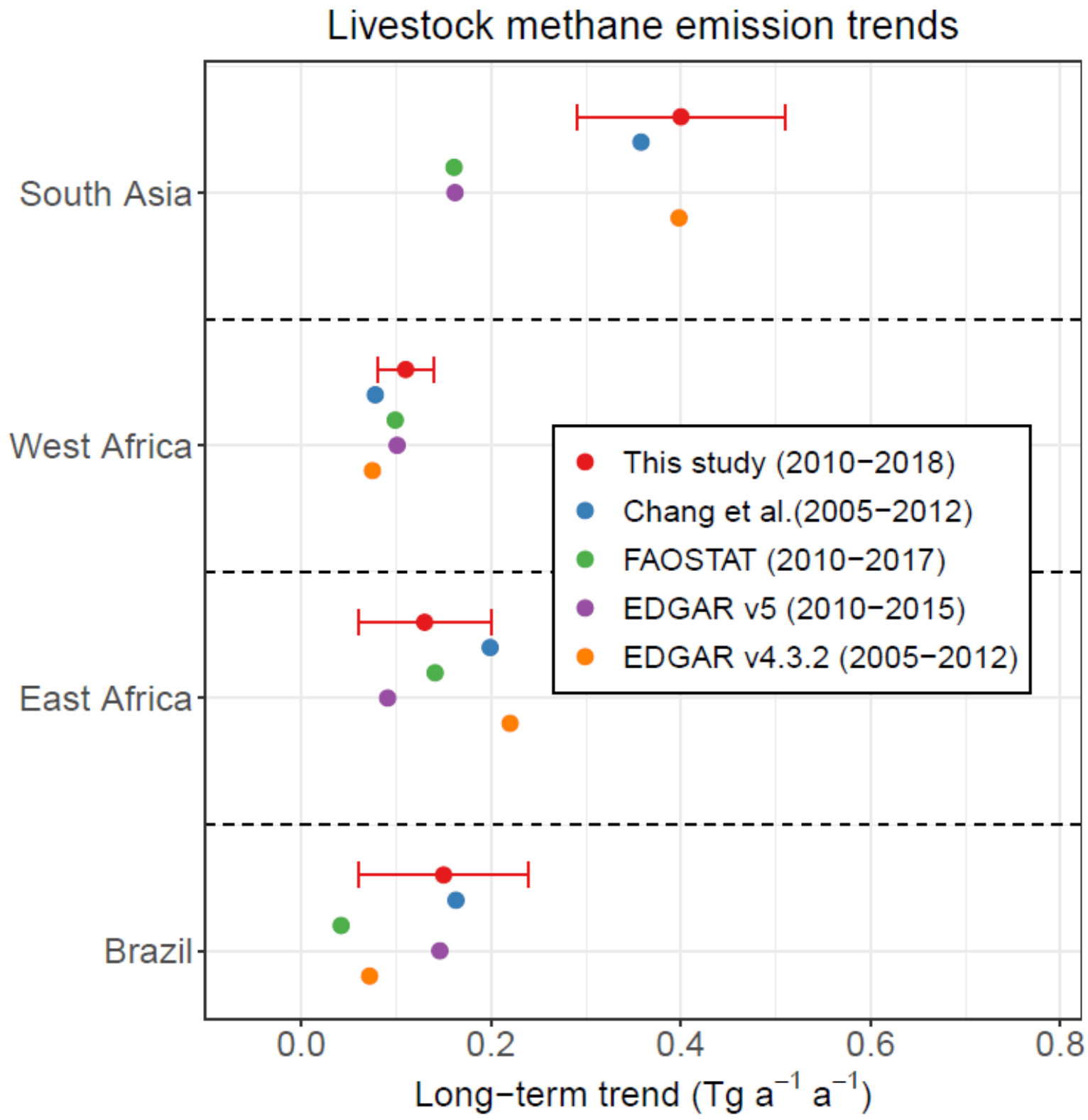

**Figure 8.** Regional trends in anthropogenic methane emissions from livestock. Our GOSAT inversion results for 2010-2018 (with error standard deviations) are compared to estimates from different bottom-up inventories over the 2005-2017 period: Chang et al. (2019), FAOSTAT (2020), EDGAR v5 (Crippa et al., 2019), and EDGAR v4.3.2 (Janssens-Maenhout et al., 2017). Results are shown for South Asia (India and Pakistan), West Africa (Nigeria, Côte d'Ivoire, Mali, Niger, Burkina Faso, Cameroon, Ghana, and Benin), East Africa (Ethiopia, Kenya, Uganda, and Tanzania), and Brazil. Our inversion assumes no prior trend in anthropogenic emissions, thus the inferred trends are solely from the GOSAT observations.

The inversion does not find significant 2010–2018 trends in anthropogenic emissions over the US (-0.12±0.21 Tg $a^{-1}$ $a^{-1}$, -0.4% $a^{-1}$). This is consistent with the lack of trend of US emissions over 2000-2014 in inversions collected by the Global Carbon Project (Bruhwiler et al., 2017). It contradicts the 2-3% $a^{-1}$ positive trend inferred from direct analysis of GOSAT enhancements (Turner et al., 2016; Sheng et al., 2018a) and the inference of large positive trends based on increasing concentrations of ethane and propane (Franco et al., 2016;Hausmann et al., 2016;Helmig et al., 2016). Bruhwiler et al. (2017) pointed out that the inference of methane emission trends from simple analysis of GOSAT data could be subject to various biases including variability in background and seasonal sampling, which would be accounted for in an inversion. Increasing ethane/methane and propane/methane emission ratios may confound inference of methane emission trends from ethane and propane records (Lan et al., 2019).

Small negative trends are found in Central Asia (Uzbekistan, Kazakhstan, Turkmenistan, Afghanistan; -0.20±0.16 Tg $a^{-1}$ $a^{-1}$), Europe (-0.15±0.15 Tg $a^{-1}$ $a^{-1}$), Australia (-0.12±0.07 Tg $a^{-1}$ $a^{-1}$). The decrease in Central Asia is attributed mainly to oil/gas, and the decrease in Australia to coal mining and livestock. Trends over Europe and Russia are too diffuse to be separated by sectors. No significant trend is found in Russia (-0.08±0.25 Tg $a^{-1}$ $a^{-1}$).

### 3.3 Wetland emissions

We infer from the inversion monthly wetland emissions for 14 subcontinental regions (Figure 1) and for individual years from 2010 to 2018, thus allowing for seasonal and interannual variability to be optimized. This achieves 167 independent pieces of information (DOFS) for wetland emissions. The results are presented as mean seasonal cycles (Figure 9) and time series of annual means (Figure 10). We find lower wetland emissions than the mean of the WetCHARTs ensemble (prior estimate) over the Amazon, the US, and Canada. The previous inversion of GOSAT data by Maasakkers et al. (2019) also found overestimation of emissions by WetCHARTs in the coastal US and Canada wetlands, but did not have significant corrections over the Amazon. The overestimation of wetland emissions in the US and eastern Canada is also reported in analyses of aircraft measurements in the southeast US (Sheng et al., 2018b) and surface observations in Canada (Baray et al., 2019), both of which used WetCHARTs v1.0 as prior information. The downregulation of North American emissions is consistent with recent WetCHARTs updates (v1.2.1) represent substantially reduced methane emissions across regions categorized as partial wetland complexes (Lehner & Döll, 2004; Bloom et al., 2017). Recent studies found that WetCHARTs overestimates wetland emissions in the Congo Basin but underestimates in the Sudd region (Lunt et al., 2019; Parker et al., 2020b; Pandey et al., 2020). Our inversion is unable to resolve this spatial correction pattern, because of coarse resolution in the wetland state vector (both regions are in Sub-Sahara Africa, i.e., wetland region 10 in Figure 1).

The seasonal cycles inferred from the inversion are in general consistent with prior information (Figure 9), although averaging kernel sensitivities indicate that we only have limited constraints on the seasonality, particularly for high latitude regions in northern hemisphere winter. This was generally expected, given the limited GOSAT observational coverage at high latitudes during winter months. The inversion infers a sharp and late (May-June) onset of methane emissions across boreal wetlands, in contrast to an early and gradual increase starting from March predicted by the prior inventory. This feature in posterior estimates appears to be consistent with aircraft and surface observations over Canada's Hudson Bay Lowlands (Pickett-Heaps et al., 2011) and eddy flux measurements over Alaskan Arctic tundra (Zona et al., 2015), while the gradual onset in the prior inventory agrees with aircraft observations over Alaska by Miller et al. (2016). The negative fluxes right before the onset may reflect strong soil sinks during spring thaw over these high-latitude regions (Jørgensen et al., 2015); alternatively, given the low averaging kernel sensitivities throughout the winter season, the negative fluxes could be attributable to spatial or temporal compensating inversion errors. The inversion also suggests stronger seasonal cycles than the prior inventory in Sub-Saharan

Africa and tropical South Asia, which indicates that the prior inventory may have underestimated the sensitivity of wetland emissions to precipitation but overestimate the sensitivity to temperature.

Our posterior estimates of 2010–2018 trends in wetland emissions are generally larger than the prior estimates from WetCHARTs, indicating that they are mostly driven by the GOSAT data. They vary greatly by region (Figure 10). Large positive trends are found in the tropics (Amazon: +0.9 Tg $a^{-1}$ $a^{-1}$; Sub-Sahara Africa: +0.6 Tg $a^{-1}$ $a^{-1}$; southern Africa: +0.4 Tg $a^{-1}$ $a^{-1}$) and high latitudes (Russia: +0.6 Tg $a^{-1}$ $a^{-1}$; West Europe: +0.4 Tg $a^{-1}$ $a^{-1}$). Increasing Amazonian wetland emissions may have been driven by intensification of flooding in the region over the past three decades (Barichivich et al., 2018). Our result of increasing tropical Africa wetland emissions is consistent with a previous regional analysis of GOSAT data, which found a positive trend of 1.5–2.1 Tg $a^{-1}$ $a^{-1}$ in the region for 2010–2016 attributed mainly to wetlands particularly the Sudd in South Sudan (Lunt et al., 2019). Compared to steady and linear increases in the tropics, boreal Russia and West Europe show abrupt increases in 2017–2018, for reasons unclear (Figure 10). Decreasing but weaker trends are found in tropical Southeast Asia (-0.3 Tg $a^{-1}$ $a^{-1}$) and Australia (-0.2 Tg $a^{-1}$ $a^{-1}$). These trends are in general not captured by prior information, suggesting that our results can be useful inputs to improvement of process-based wetland emission modeling. Furthermore, posterior inter-annual variability is overall larger than prior variability across all regions, suggesting that the integrated climate sensitivity of prior emissions may be underestimated.

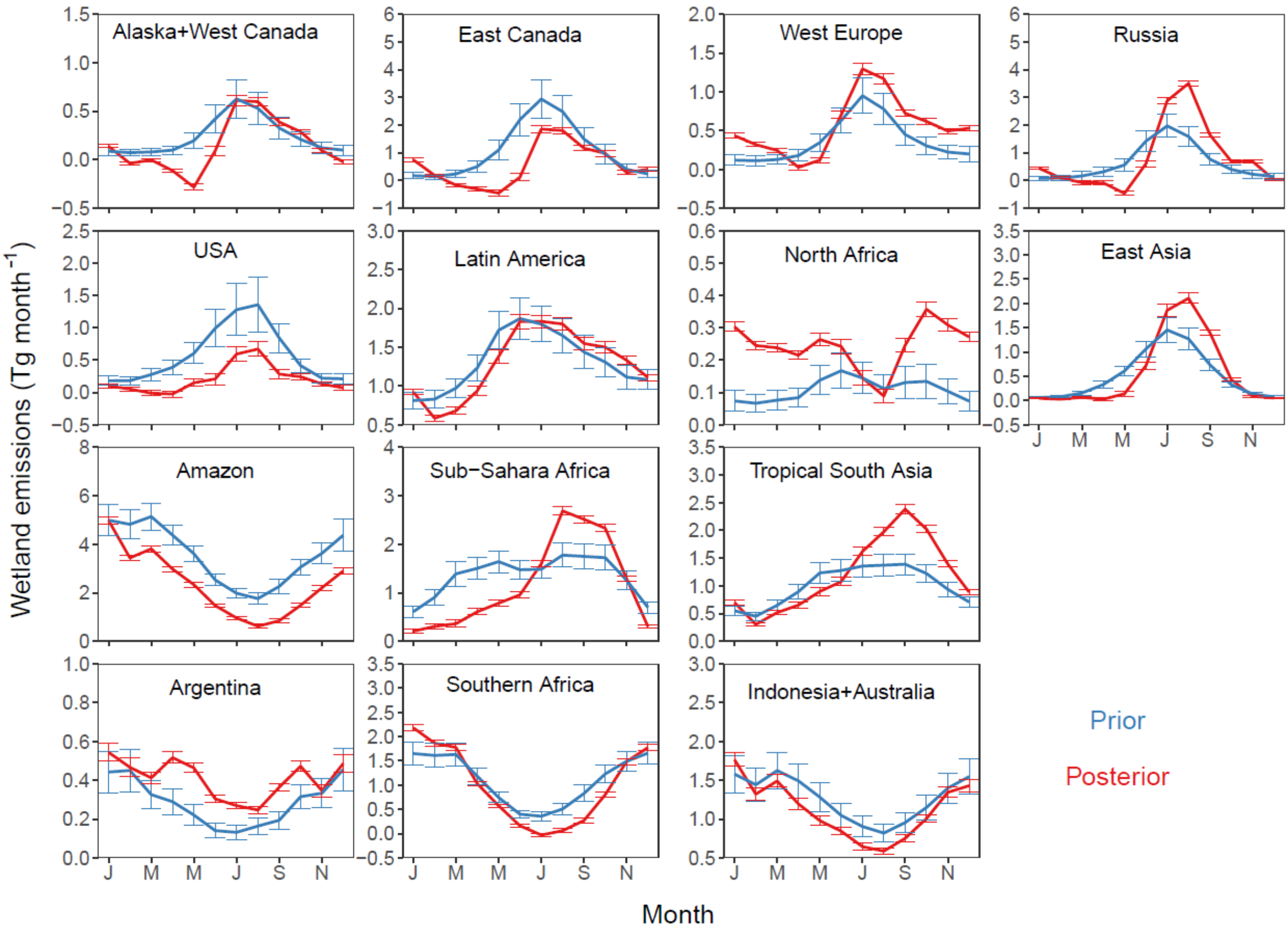

455 **Figure 9.** Seasonal variation of wetland emissions for 14 subcontinental regions (Figure 1). Values are means for 2010–2018. The prior estimate is the mean of the WetCHARTs inventory ensemble (Bloom et al., 2017). The posterior estimate is from our inversion of GOSAT data. Vertical bars are error standard deviations. The reduction of error in the posterior estimate measures the information content provided by the GOSAT data. Scales are different between panels.

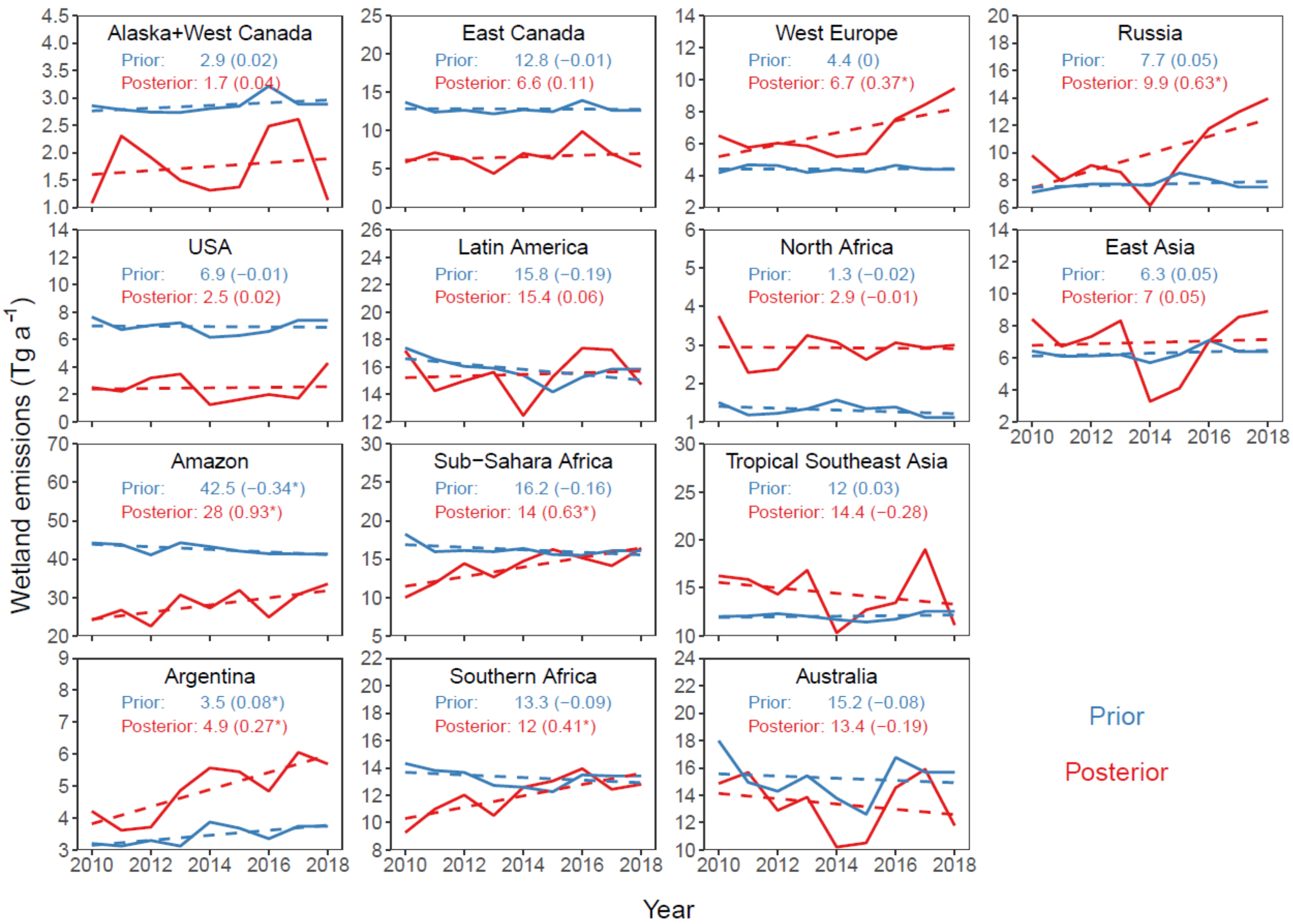

**Figure 10.** Wetland emission trends, 2010–2018. The figure shows annual mean emissions for the prior estimate (mean of WetCHARTs inventory ensemble) and the posterior estimate after inversion of GOSAT data. Values are for the 14 subcontinental regions of Figure 1. The trends are from ordinarly linear regression. Inset are prior and posterior 2010–2018 average annual emissions in Tg $a^{-1}$ with 2010–2018 trends in Tg $a^{-1}$ $a^{-1}$ in parentheses. Significant trends at the 95% confidence level are denoted with "*". Note differences in scales between panels.

### 3.4 OH concentration

Our inversion infers a global methane lifetime against oxidation by tropospheric OH of 12.4±0.3 a, which is ~ 15% longer than the prior estimate (10.7±1.1 a) and is at the higher end of the inference from the methyl chloroform proxy (11.2±1.3 years) (Prather et al., 2012). We find that the downward correction for OH concentrations is mainly in the northern hemisphere. The north-to-south inter-hemispheric OH ratio is 1.02±0.05 in the posterior estimate, as compared to 1.16 in the prior estimate and 1.28±0.10 in the ACCMIP model ensemble (Naik et al., 2013). It is more consistent with the observation-based estimate of 0.97 ± 0.12 (Patra et al., 2014). No significant 2010–2018 trend is seen in the methane lifetime (Figure 11). The OH

concentration is 5% lower than average in 2014, which had particularly large peatland fires in Indonesia that would be very large sources of carbon monoxide (CO) as a sink for OH (Pandey et al., 2017b).

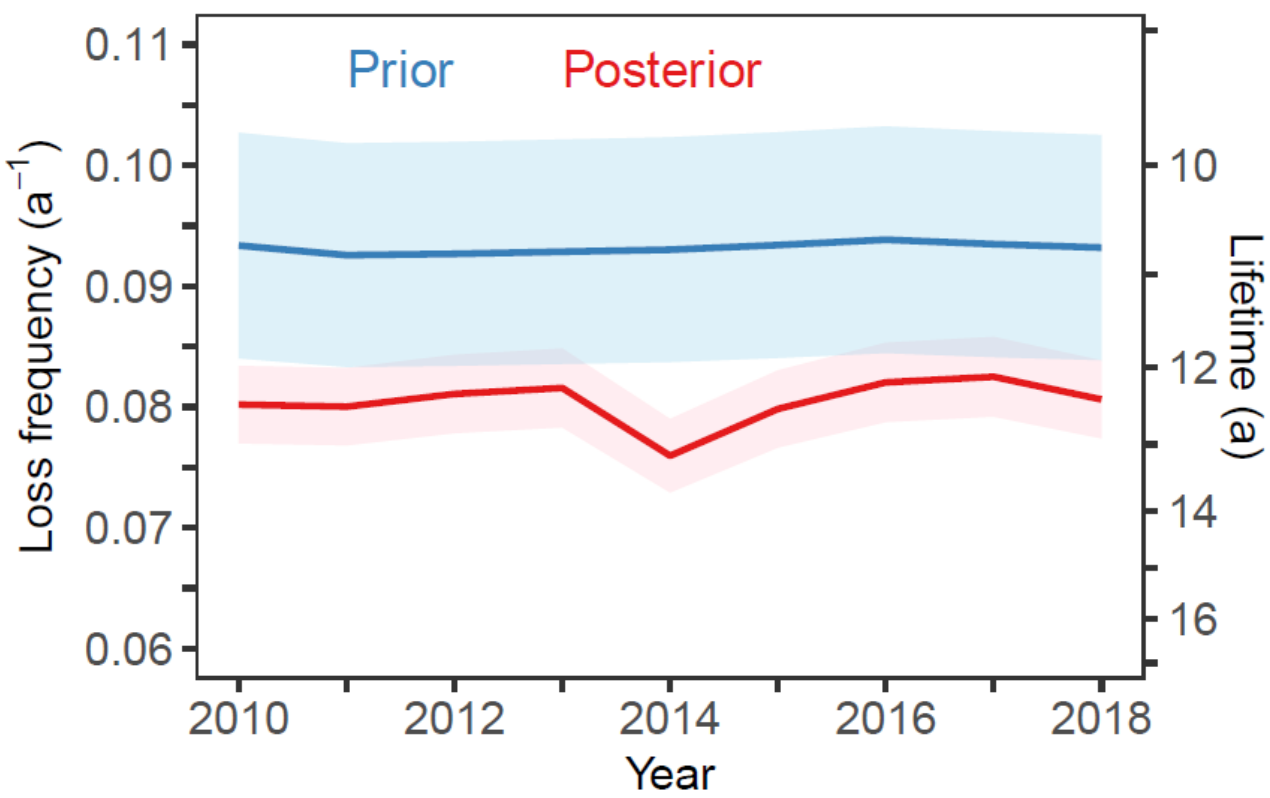

**Figure 11.** Methane loss frequency and lifetime against oxidation by tropospheric OH, 2010-2018. Values are annual means with error standard deviations. The loss frequency ($k$) is as calculated by Eq. 1 and the lifetime ($\tau$) is the inverse. The prior estimate from Wecht et al. (2014) assumes no 2010–2018 trend in OH concentrations; the slight variability seen in the Figure is due to temperature.

### 3.5 Attribution of the 2010–2018 methane trend

Figure 12 shows the 2010–2018 annual methane growth rates inferred from NOAA background surface measurements (Dlugokencky, 2020) and from our inversion of GOSAT data. There is general consistency between the two, with both showing highest growth rates in 2014–2015 and a general acceleration after 2014. Our inversion achieves a much better match to the interannual variability in the NOAA record than the previous work of Maasakkers et al. (2019), in large part because of our optimization of the interannual variability of wetland emissions.

The bottom panel of Figure 12 attributes the interannual variability in the methane growth rate to individual processes as determined by the inversion. The growth rate $G_j = [dm/dt]_j$ in year $j$, where $m$ is the global methane mass, is determined by the balance between sources and sinks:

$$G_j = E_j - k_j m_j - L_j \tag{9}$$

Here $E_j$ denotes the global emission in year $j$, for which the inversion provides further sectoral breakdown, $k_j$ is the loss frequency against oxidation by tropospheric OH (Eq. 1), $m_j$ is the total methane mass, and $L_j$ represents the minor sinks not optimized by the inversion. Let $\Delta E_j = E_j - E_o$, $\Delta k_j = k_j - k_o$, and $\Delta m_j = m_j - m_o$ represent the changes relative to 2010 conditions ($E_o$, $k_o$, $m_o$) taken as reference. We can then write

$$G_j = (E_o + \Delta E_j) - (k_o + \Delta k_j)(m_o + \Delta m_j) - (L_o + \Delta L_j)$$
$$\approx (E_o - k_o m_o - L_o - k_o \Delta m_j) + \Delta E_j - m_o \Delta k_j \quad (10)$$

where we have neglected the minor terms $\Delta k_j \Delta m_j$ and $\Delta L_j$. Here the growth rate $G_j$ in year $j$ is decomposed into three terms: (1) a relaxation to steady state based on 2010 conditions ($E_o - k_o m_o\ L_o - k_o \Delta m_j$), (2) a perturbation to emissions ($\Delta E_j$) that can be further disaggregated by sectors, and (3) a perturbation to OH concentrations ($m_o \Delta k_j$).

We see that from the bottom panel of Figure 12 that the legacy of 2010 imbalance sustains a positive growth rate throughout the 2010–2018 period but this influence diminishes towards the end of the record. The 2010–2018 trend in anthropogenic emissions is linear by design of the inversion and sustains a steady growth rate over the 2010–2018 period as the legacy of the 2010 imbalance declines. The spike in the methane growth rate in 2014–2015 is attributed to an anomalously low OH concentration in 2014 (5% lower than 2010–2018 average; Figure 11), partly offset by low wetland emissions, and anomalously high fire emissions in 2015, mostly from Indonesia peatlands (Worden et al., 2017). Smoldering peatland fires are particularly large sources of methane (Liu et al., 2020). The large fire emissions are informed by the GFED inventory because the inter-annual variability of fire emissions is not optimized in our inversion. Despite their small magnitude relative to wetland and anthropogenic emissions globally, anomalous fire emissions can be an important contributor to methane interannual variability (Worden et al., 2017;Pandey et al., 2017b), both directly by releasing methane and indirectly by decreasing OH concentrations through CO emissions. Error correlations in our methane trend attributions (see discussion below) suggest that the high growth rates in both 2014 and 2015 could be due to Indonesian fires including effects on OH.

In addition to the 2014–2015 extremum, the NOAA surface observations show an acceleration of methane growth during the latter part of the 2010–2018 record (Nisbet et al., 2019) and this is reproduced in our inversion (Figure 12). We find that the rapid growth in 2016–2018 is mostly driven by wetlands, including contributions from both the steady 2010–2018 increase in tropical wetlands (in particular the Amazon and tropical Africa) and the 2016–2018 surge in Eurasian boreal wetlands (Figure 10).

Figure 13 shows the mean 2010–2018 emission trends attributed by the inversion to individual sectors. The 2010–2018 growth in emissions is two-thirds from wetlands (3.0 Tg $a^{-1}$ $a^{-1}$) and one third anthropogenic (1.5 Tg $a^{-1}$ $a^{-1}$). Wetland emissions increase in both tropical regions (1.8±0.6 Tg $a^{-1}$ $a^{-1}$; Amazon and tropical Africa) and the extra-tropics (1.2±0.3 Tg $a^{-1}$ $a^{-1}$; Russia and West Europe) (Figure 13 and Figure 10). The increase of anthropogenic emissions is driven by livestock (South Asia, tropical Africa, Brazil), rice (East Asia), and wastewater treatment (Asia) sectors (Figure 13). The best estimate for global trends in emissions from fuel exploitation (oil, gas, and coal) (0.0±0.4 Tg $a^{-1}$ $a^{-1}$) is almost zero; but small trends cannot be ruled out given the uncertainty (Figure 13).

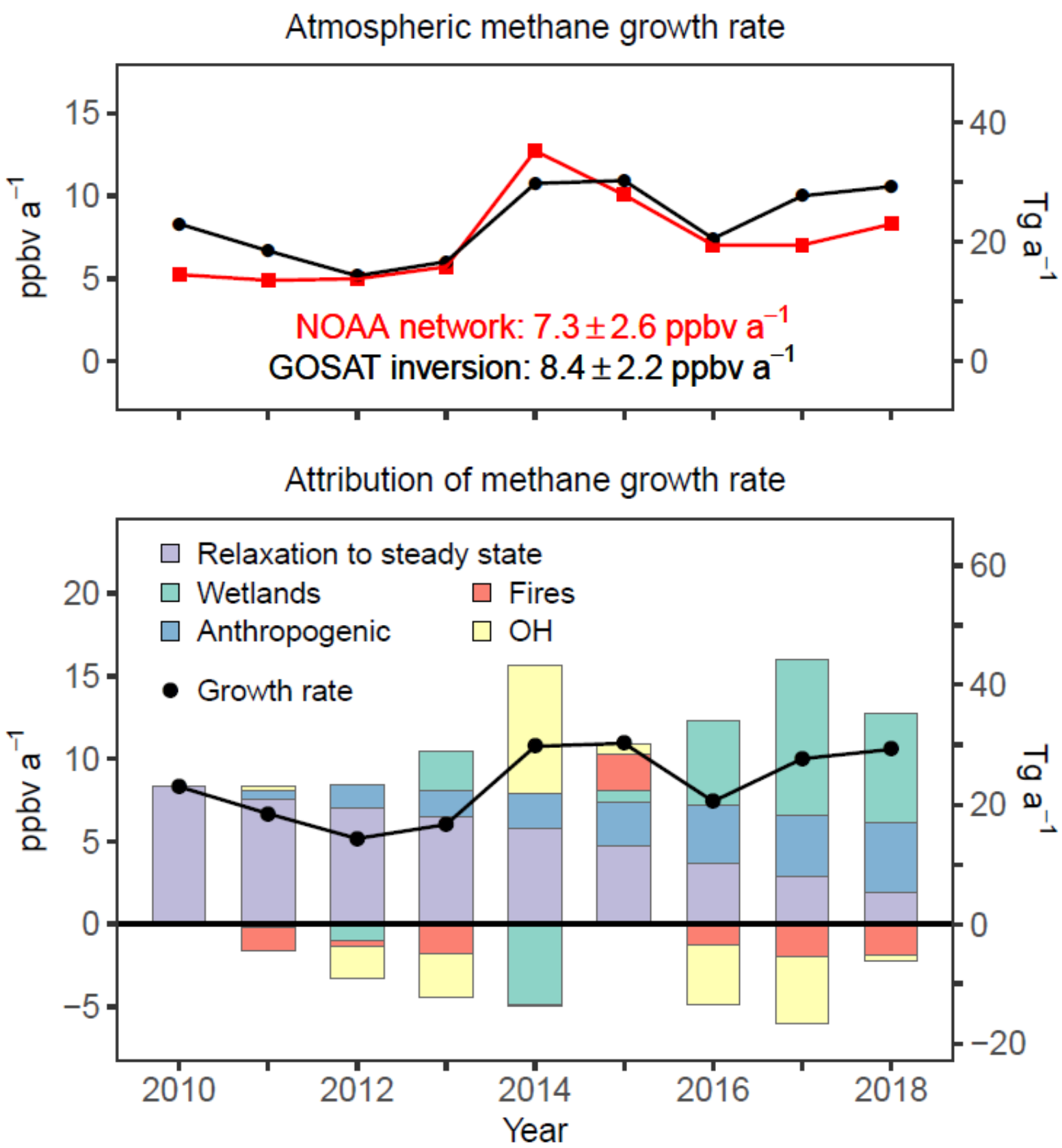

**Figure 12.** 2010–2018 annual growth rates in global atmospheric methane. (Top) Comparison of annual growth rates inferred from our inversion of GOSAT data and from the NOAA surface network (Dlugokencky, 2020). Average methane growth rates for the period are inset. (Bottom) Attribution of annual growth rates in the GOSAT inversion to perturbations to emissions (anthropogenic, wetlands, fires) and OH concentrations relative to 2010 conditions. The purple bar shows the relaxation of 2010 budget imbalance to steady state. See text for details explaining the breakdown.

We estimate from the inversion a global mean methane emission for 2010–2018 of 510±4 Tg $a^{-1}$ (wetlands: 139 Tg $a^{-1}$; anthropogenic: 341 Tg $a^{-1}$) and a sink of 488±4 Tg $a^{-1}$. This posterior global emission is lower than the prior estimate (538 Tg $a^{-1}$) and the 538–593 Tg $a^{-1}$ range reported recently by the Global Carbon Project for 2008–2017 (Saunois et al., 2020). Compared to prior emissions, we estimate lower emissions for wetlands and fossil fuel, and higher emissions for livestock and rice (Figure 13). Meanwhile, we estimate a methane lifetime against tropospheric OH oxidation of 12.4±0.3 years, at the high end of 11.2±1.3 years based on the methyl chloroform proxy (Prather et al., 2012).

Figure 14 plots the posterior joint probability density functions (PDFs) for pairs of global budget terms and their trends. There is a strong negative error correlation in the inversion results between anthropogenic emissions and methane lifetime (r=-0.8), reflecting limited ability of the inversion to separate the two. In contrast, error correlations between wetland emissions and methane lifetime (r=-0.4), and between wetland and anthropogenic emissions (r=-0.2) are much smaller. We find moderate error correlations between the OH trend and either wetland or anthropogenic emission trends (r=-0.6). We cannot exclude at the 90% confidence level the possibility that the 2010–2018 anthropogenic emission trend could be zero (compensated by a decrease in OH concentrations), but we can exclude the possibility that the 2010–2018 wetland emission trend could be zero. Improved separation of global budget terms and their trends may be achieved by including additional information from surface observation (Lu et al., 2020) and from thermal infrared satellite observations (Y. Zhang et al., 2018).

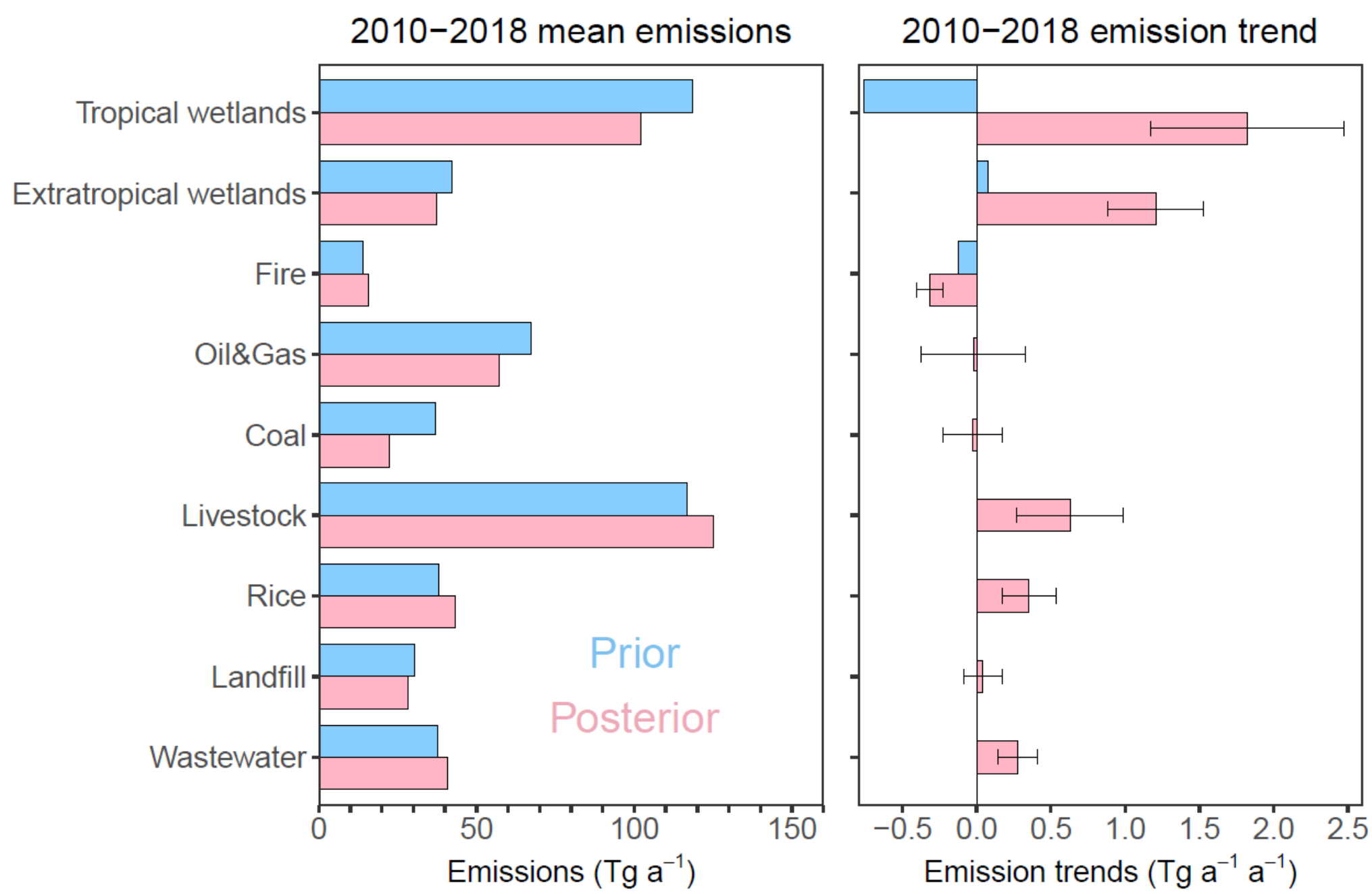

**Figure 13.** 2010–2018 global methane emissions and emission trends partitioned by individual sectors. Posterior estimates are from our inversion of GOSAT data. Prior estimates for anthropogenic emission trends are zero. Error bars in the right panel shows posterior error standard deviations for emission trends. Posterior error standard deviations are too small to show for mean emissions of the left panel. Posterior errors computed from Eq. 4 and 7 tend to be overoptimistic because of ideal inversion assumptions (Maasakkers et al., 2019).

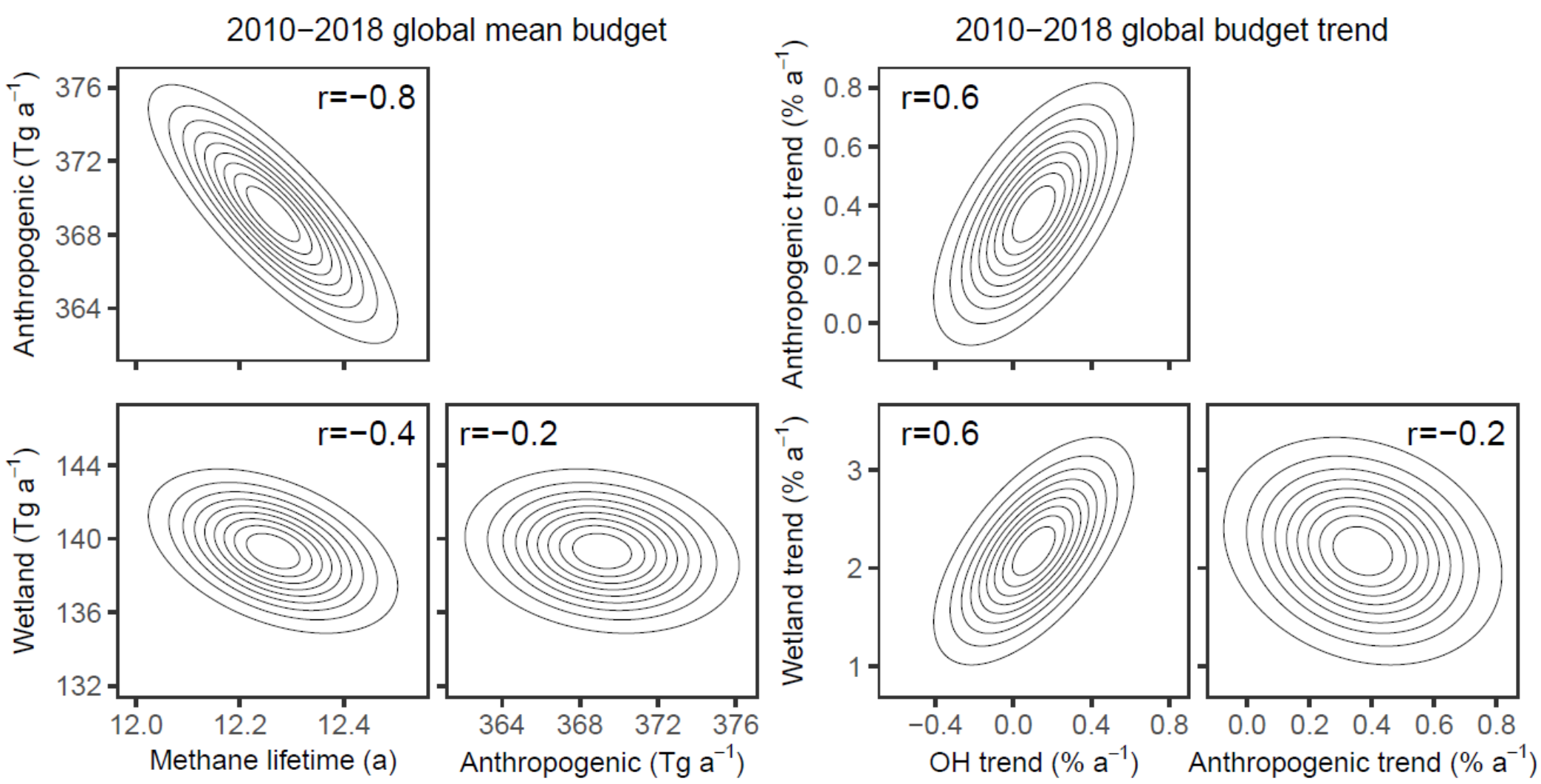

**Figure 14.** Error correlations between global anthropogenic emissions, wetland emissions, and tropospheric OH concentrations (methane lifetime against oxidation by tropospheric OH; $\tau$) in the inverse solution. Results are shown both for 2010–2018 mean values and for 2010–2018 trends. The error correlations are presented as joint probability density functions for pairs of reduced global state vector elements. Confidence ellipses represent probability of 0.1 (innermost) to 0.9 (outermost) at intervals of 0.1. The error correlation coefficients are shown inset.

## 4 Conclusions

We quantified the regional and sectoral contributions to global atmospheric methane and its 2010–2018 trend by inversion of GOSAT satellite observations. The inversion jointly optimizes (1) 2010–2018 anthropogenic emissions and their linear trends on a 4°×5 ° grid; (2) monthly wetland emissions in 14 subcontinental regions for individual years; and (3) annual mean hemispheric OH concentrations for individual years. Analytical solution to the optimization problem provides closed-form estimates of posterior error covariances and information content, allowing us in particular to diagnose error correlations in our solution. Separate optimization of wetland and anthropogenic emissions allows us to resolve interannual and seasonal variations in posterior wetland emissions. Our inversion introduces additional innovations including the correction of stratospheric model biases using ACE-FTS satellite data, and a new bottom-up inventory for emissions from fossil fuel exploitation based on national reports to the UNFCCC (Scarpelli et al., 2020).

Our optimization of 2010–2018 mean anthropogenic emissions on the 4°×5 ° grid provides strong information in source regions as measured by averaging kernel sensitivities. We find that estimates of anthropogenic emissions reported by individual countries to the UNFCCC are too high for China (coal emissions) and Russia (oil/gas emissions) and too low for Venezuela (oil/gas) and the U.S. (oil/gas). We also find that tropical livestock emissions are larger than previous estimates particularly in South Asia, Africa, and Brazil. Our posterior estimate of anthropogenic emissions in India (33 Tg $a^{-1}$) is much higher than its most recent (2010) report to the UNFCCC (20 Tg $a^{-1}$), mostly because of livestock emissions.

2010–2018 trends in methane emissions on the 4°×5 ° grid are successfully quantified in source regions. We find that the largest growth in anthropogenic emissions is from tropical livestock in South Asia, tropical Africa, and Brazil. This finding is consistent with trends in livestock populations. There has been little discussion in the literature about increasing agricultural methane emissions in these developing countries (Jackson et al., 2020). The 2010–2018 increase in Chinese emissions is smaller than previously reported in inversions focused on earlier periods, likely caused by leveling of coal emissions in China. The 2010–2018 emission trend in the US is insignificant on the national scale.

We find that global wetland emissions are lower than the prior estimate from mean WetCHARTs emissions, mostly because of the Amazon. Wetland emissions over North America are also lower, consistent with previous studies. In both cases, we note that posterior estimates are all well within the full WetCHARTs uncertainty range (Bloom et al., 2017). The seasonality of wetland emissions inferred by the inversion is in general consistent with WetCHARTs. An exception is in boreal wetlands where we find negative fluxes in April-May, possibly reflecting uptake as the soil thaws. The inversion infers increasing wetland emissions over the 2010–2018 period, superimposed on large inter-annual variability, in both the tropics (Amazon, tropical Africa) and extra-tropics (Russia, western Europe).

Our optimization of annual hemispheric OH concentrations yields a global methane lifetime of 12.4±0.3 years against oxidation by tropospheric OH, with an inter-hemispheric OH ratio of 1.02. Our best estimate is that the global OH concentration has no significant trend over 2010–2018 except for a 5% dip in 2014.

Taking all these methane budget terms together, our inversion of GOSAT satellite data estimates global mean methane emissions for 2010–2018 of 510 Tg $a^{-1}$, with 341 Tg $a^{-1}$ from anthropogenic sources, 139 Tg $a^{-1}$ from wetland sources, and 30 Tg $a^{-1}$ from other natural sources. Our inferred growth rate of methane over that period matches that observed at NOAA background sites, including peak growth rates in 2014–2015 and an overall acceleration over the 2010–2018 period. We attribute the 2014–2015 peaks in methane growth rates to low OH concentrations (2014) and high fire emissions (2015), and the overall trend acceleration to a sustained increase in emissions. Most of this increase in emissions is attributed to wetlands (tropics: 1.8 Tg $a^{-1}$ $a^{-1}$; extra-tropics: 1.2 Tg $a^{-1}$ $a^{-1}$) and tropical livestock (0.8 Tg $a^{-1}$ $a^{-1}$). Our best estimate indicates no contribution of the oil/gas sector to increasing global emissions; but small oil/gas trends cannot be ruled out given relatively large uncertainties. Our finding is in general consistent with a previous 2010–2015 inversion of GOSAT data (Maasakkers et al., 2019) although here we use a longer record and we capture the interannual variability better. Our results also agree with isotopic data indicating that the rise in methane is driven by biogenic sources (Schaefer et al., 2016;Nisbet et al., 2016). The increase in tropical livestock emissions is quantitatively consistent with bottom-up estimates. More work is needed to understand the increase in wetland emissions.

**Data availability**

The GOSAT proxy satellite methane observations are available at the CEDA archive (Parker and Boesch, 2020). The ACE-FTS satellite observations can be requested through http://www.ace.uwaterloo.ca/data.php (last access: July 20, 2020). TCCON data were obtained from the TCCON Data Archive hosted by CaltechDATA (https://tccondata.org) (Deutscher et al., 2017; Dubey et al., 2017; Feist et al., 2017; Goo et al., 2017; Griffith et al., 2017a, b; Hase et al., 2017; Iraci et al., 2017a, b; Kivi et al., 2017; Liu et al., 2017; de Maziere et al., 2017; Morino et al., 2017a, b, c; Notholt et al., 2019a, b; Sherlock et al., 2017a, b; Shiomi et al., 2017; Strong et al., 2017; Sussmann et al., 2017; Te et al., 2017; Warneke et al., 2017; Wennberg et al., 2017a, b, c, d; Wunch et al., 2017). NOAA surface observations are accessed through NOAA ESRL/GMD CCGG Group (doi.org/10.15138/VNCZ-M766) (Dlugokencky et al., 2020). National reports to UNFCCC are available through UNFCCC's Greenhouse Gas Inventory Data Interface (di.unfccc.int/detailed_data_by_party, last access: July 20, 2020). EDGAR anthropogenic emission inventories (v4.3.2 and v5) are available at https://data.europa.eu/doi/10.2904/JRC_DATASET_EDGAR (last access: July 20, 2020).

**Author contributions**

YZ and DJJ designed the study. YZ conducted the modeling and data analyses with contributions from XL, JDM, TRS, MPS, JXS, LS, ZQ. RJP and HB provided the GOSAT satellite methane retrievals. AAB and SM contributed to the WetCHARTs wetland emission inventory and its interpretation. JFC contributed to analyses and interpretation of bottom-up livestock emission inventories. YZ and DJJ wrote the paper with inputs from all authors.

**Competing interests**

The authors declare that they have no conflict of interest.

**Acknowledgments**

Work at Harvard was supported by the NASA Carbon Monitoring System (CMS), Interdisciplinary Science (IDS), and Advanced Information Systems Technology (AIST) programs. YZ was supported by the foundation of Westlake University, the Kravis Fellowship through the Environmental Defense Fund (EDF), and Harvard University. YZ thanks Peter Bernath and Chris Boone for discussion on the ACE-FTS data. Part of this research was carried out at the Jet Propulsion Laboratory, California Institute of Technology, under a contract with NASA. RJP and HB are funded via the UK National Centre for Earth Observation (NE/R016518/1 and NE/N018079/1). RJP and HB also acknowledge funding from the ESA GHG-CCI and Copernicus C3S projects. We thank the Japanese Aerospace Exploration Agency, National Institute for Environmental Studies,

and the Ministry of Environment for the GOSAT data and their continuous support as part of the Joint Research Agreement. GOSAT retrievals were performed with the ALICE High Performance Computing Facility at the University of Leicester.

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
