# Peer review of "Attribution of the accelerating increase in atmospheric methane during 2010–2018 by inverse analysis of GOSAT observations"

_Atmospheric Chemistry and Physics, 2020_

## Referee Comment (RC1) · Anonymous Referee #1 · 23 Oct 2020

Zhang et al uses an analytical inverse method to estimate methane emissions from the GOSAT data from 2010 to 2018. The study contains a lot of information that is poorly structured and explained so I found it difficult to extract any new or notable scientific insights. I recommend a major revision that takes into the recommendations I have made below to help draw out the key points of the paper. Given the eminence of the coauthors I feel slightly aggrieved I have needed to make some of these points.

Minor Comment

I think the language is a bit odd in places. Easily fixed but needs an overhaul.

Major Clarifications

[Figure]

Cleanly separating wetland and non-wetland emissions, including rice paddies, etc is a bold claim. The authors' motivation, based on WetCHARTs, is that wetlands have relatively coherent spatial behaviours. From what I understand the authors' state vector is at grid-point resolution for non-wetland sources and their trend, but the wetland emissions are described on much larger spatial regions. I remain unclear whether using a combination of small and large geographical regions will decrease or increase posterior correlations between wetlands and anthropogenic emissions. An argument could be made for both sides. Certainly, more discussion/description is needed.

These two statements are apparently contradictory: Line 155: "Our prior estimate assumes no 2010-2018 trends in non-wetland emissions...." Line 160: "We specify an absolute error standard deviations of 5%/a for linear trends of non-wetland emissions..."

Line 161: Given the diversity of emission estimates from the ensemble members of WetCHARTs and the resulting uncertainty covariance structure, I am left wondering how sensitive the authors' posterior solution is to this information.

Line 221: Given your reliance on the residual error method I think it would be useful to explain this a bit more.

Line 224: Please clarify whether you use the brute force method to calculate the Jacobian matrix or take advantage of the tagged CH4 simulation, as described in section 2.4.

Line 255: Comparison with NOAA is an important first step to determine whether your posterior solution is consistent with the observed atmospheric growth rate. Unfortunately, Figure 3 and the accompanying text is not sufficiently clear for this reviewer to make that judgement. For the lower right panel of Figure 3 it would be better to use a smaller y axis range given the small differences. At least then a reader can eyeball the comparison. I recommend "better fit" is substituted with some quantitative statistics, e.g. bias, correlations, etc.

[Figure]

Comment on arrangement of paper: showing your ability to independently estimate wetland and non-wetland emissions on the spatial scale you are using (Figure 14) would be useful up front before the more detailed discussions begin. See my comments below.

A broad comment relevant to more than one figure, e.g. Figure 3: the period 2010 to 2018 covers a wide range of climate variations (e.g. both phases of ENSO) so that taking the mean or differences over this period hides a lot of useful information about how methane has changed.

Line 289: averaging kernel sensitivities? The matrix of averaging kernels already describes sensitivity. Do the authors mean the sensitivities described by this matrix?

Line 292: from what I understand the state vector has a length of 1000s but the DOFs is limited to 179. What implications does that have for being able to resolve the state vector?

Line 291: "By applying the posterior/prior correction factors to the prior distribution of each anthropogenic emission sector, we obtain improved estimates for anthropogenic emissions for that sector." This statement does not make sense. Improved how? Better fit to measurements? Smaller uncertainties? Applying correction factors does not improve estimates for anthropogenic emissions. Which sector?

Comments on reporting results: some posterior estimates are accompanied by their uncertainties and some are not. The authors need to be consistent. They should certainly report them on Figure 6.

Line 315: I am really interested to hear more about how their results point towards an underestimate in livestock emissions. This whole paragraph contains so much hand waving I thought I was at a pre-pandemic music festival. I urge the authors to justify their result in light of it being inconsistent with Maasakkers et al and other published studies that attribute most of these changes to wetlands. I acknowledge they take a

second swing at this point on page 15 but I was unsure which they used for their prior (for Figure 8) and more importantly how confidently they could separate wetland and livestock emissions.

Line 358: why are the constraints strongest over India and China?

Line 539: That's a very long lifetime for methane against OH oxidation. This reviewer is left wondering whether the authors' inverse problem remains ill-posed given their state vector. The joint PDF clearly shows strong correlations between different parts of the (global mean) state vector. Given the authors reporting of regional methane missions, it would be useful to show the regional joint PDFs for which I suspect the correlations between anthropogenic and wetland emissions can be much higher. Their conclusion on page 24 doesn't fill me with confidence.
* * *

---

## Referee Comment (RC2) · Anonymous Referee #2 · 14 Nov 2020

This article analyses the role of wetland and livestock on the global CH4 growth rates in the period 2010-2018. The authors have also carefully assesed the role of OH on the CH4 growth rate. The article is generally well written, and has built upon their earlier analysis. I have doubt on the simplicity of the emission optimisation for only two major sectors, while CH4 has so many other natural and anthropogenic emission sectors those vary very differently at interannual and longer timescales. Detailed comments below. The manuscript may be considered for publication after addressing some of these comments.

Specific comments: lines 38-39 : need references?

[Figure]

lines 76ff: I find later that the CTM configuration and other details are discussed later. May be remove this paragraph !!

lines 90ff: I think this is not an ideal choice because the values of CH4 depends on assumed CO2 concentrations. We know that our understanding of CO2 fluxes over many data void regions are poor. How do you estimate regional bias, say over South Asia, Southeast Asia, Amazon/Brazil, Africa etc.

line: that's the lower bound, what about the oceans?

line 137: Did you apply any scaling to termite emissions?

lines 170ff: 1-year spin-up is too short by any standards for long-lived species. What about the stabilisation of vertical gradients? You cannot get that right by the unbiasing method you describe. For future studies please make a spin-up for 10 years or so. Its worth the computing cost, given the large amount of follow work and analysis is done for any publication.

Figure 2 and associated text: Does these correction factors extend till the poles? The problem is that the stratosphere is dynamic as well. Are such seasonal bias correction factor good for getting the profiles right.

My worry here is also that if the stratosphere is not spun-up well these bias correction factors will not be time invariant. So the trends in anthropogenic or natural emissions you derive later may not be free of these correction factors.

Figure 5 and associated text: Very tough to accept the results for the a posteriori patterns, e.g., there is a strip of increased emissions along the Himalayan region! Is this arising from not properly accounting for the orography in the coarse resolution model? Or this may be an artifact of the proxy retrievals by miscalulating CO2 over northern part of the Indian subcontinent.

Figure 6 and associated text: I like this analysis but it is not clear at all, if the results are independent of the priors! For example, if you if you started with the priors from the

UNFCCC will you get the same a posteriori emissions?

The question is also same for the trends in derived emissions. Some sensitivity tests are need for clarifying robustness of the a posteriori emissions.

See for example Patra et al., JMSJ, 2016. That paper is also relevant for other discussion in the paper where you discuss trends of emissions over China, and from Animals etc.

lines 340-344: How confidently we can talk about the oil and gas emission trends - given that the cited references are so small scale compared to the gridcell you optimize here. For example, can you gather the cited references by model gridcells and check the validity of the inversion and vice versa?

lines 366-377: I am a bit confused why/how all the anthropogenic emissions are linked with the livestock population? Is this because your a priori emissions only accounts for the livestocks? For example how is the trends in waste management in these developing nations?

Page 18-20: I do not know but I have a feeling that the inversion is set in such a way that all the regions are having somewhat similar increase in emissions, either from the animals or from wetlands. The authors are in the best position to judge and find the reasons behind such outcomes from their inversion system.

If you can explain the global CH4 growth rates using emissions from only two emission sectors then we have an oversimplifications of CH4 sources. I cannot prove anything in favour or against the proposed mechanisms here but many of the hypothesis are based on apparently single line of evidence, i.e., the GOSAT proxy XCH4 retrievals. Aren't Maasakkers et al., Lunt et al., Pandey et al., Parker et al. using the same XCH4 data sources?

In addition to the trends analysis, it would have been useful to discuss how and why are the large interannual variability in some regions as shown in Fig. 10. Are there irregular

data gaps or a particular climate anomaly affecting CH4 emissions regionally?

line 470: The NH/SH OH ratio of 1.02 is an interesting result from this exercise. Could you please comment here how are the NH/SH ratio of CH4 emission changed from a priori to a posteriori cases? I suspect the derived NH/SH OH ratio depends on how well freely the inversion system is allowed to adjust emissions vs that for OH.

page 23: How can we get assured that the OH and CH4 emissions can be optimised in one inversion system. Is there a dipole effect? For example, if you do more or less number of iterations, will the global total emission and global mean OH will be different? I see that this issue is addressed later using Fig. 14, but still not convinced that the inversion system is separating the wetland vs livestock emissions well.

Figure 13: You attributed emissions from livestock increase to the animal population, but what appears here is that the increase in emissions from tropical and extratropical wetland are the greatest. Could you propose a mechanistic viewpoint ?

---

## Author Comment (AC1) · 1 Jan 2021

We thank both reviewers for their comments and suggestions.

Reviewer #1

Zhang et al uses an analytical inverse method to estimate methane emissions from the GOSAT data from 2010 to 2018. The study contains a lot of information that is poorly structured and explained so I found it difficult to extract any new or notable scientific insights. I recommend a major revision that takes into the recommendations I have made below to help draw out the key points of the paper. Given the eminence of the coauthors I feel slightly aggrieved I have needed to make some of these points.

Thanks for your suggestions. We have worked through the manuscript to make the improvements.

Minor Comment

I think the language is a bit odd in places. Easily fixed but needs an overhaul.

We tried our best to improve the language throughout the text.

Major Clarifications

Cleanly separating wetland and non-wetland emissions, including rice paddies, etc is a bold claim. The authors' motivation, based on WetCHARTs, is that wetlands have relatively coherent spatial behaviours. From what I understand the authors' state vector is at grid-point resolution for non-wetland sources and their trend, but the wetland emissions are described on much larger spatial regions. I remain unclear whether using a combination of small and large geographical regions will decrease or increase posterior correlations between wetlands and anthropogenic emissions. An argument could be made for both sides. Certainly, more discussion/description is needed.

We now add a new figure (Figure 6), which shows the posterior error correlations between regional wetland and anthropogenic emissions. The error correlations are moderate (r: averages -0.3; range -0.1 to -0.5), suggesting that the inversion has some ability to separate the two at the regional scale.

We also test the impact of unresolved subregional wetland distribution on both wetland and anthropogenic estimates with an additional sensitivity inversion. We perturb prior wetland distribution in Africa following Lunt et al. (2019) (which reported that WetCHARTS underestimate over the Sudd region but overestimate over the Congo Basin). This results in some changes in estimates for both wetland and anthropogenic

emissions in the region, but the main conclusion still holds. The sensitivity inversion is presented in Figure S1 (prior wetland emission), Figure S4 (anthropogenic emissions), and Figure S5 (wetland emissions). The resulting impact on the inference of anthropogenic emissions in eastern Africa is also presented in Figure 8.

We also make changes in Section 2.2 (state vector) to better explain our reasoning on separate optimization of wetland and anthropogenic emissions.

These two statements are apparently contradictory: Line 155: "Our prior estimate assumes no 2010-2018 trends in non-wetland emissions. . .." Line 160: "We specify an absolute error standard deviations of 5%/a for linear trends of non-wetland emissions. . ." They are not contradictory. The first statement describes the prior estimate for non-wetland trends, and the second describes the error statistics for this prior estimate. We now revise the two sentences to improve clarification.

Line 161: Given the diversity of emission estimates from the ensemble members of WetCHARTs and the resulting uncertainty covariance structure, I am left wondering how sensitive the authors' posterior solution is to this information. We have tested the inversion without the error covariance structure (only diagonal terms) from the WetCHARTS ensemble. We find the resulting changes in the posterior solution are small. In addition, as described above, we perform a sensitivity inversion to test the impact of errors in subregional wetland distribution unresolved by our inversion.

Line 221: Given your reliance on the residual error method I think it would be useful to explain this a bit more. We now explain the method in the text.

Line 224: Please clarify whether you use the brute force method to calculate the Jacobian matrix or take advantage of the tagged CH4 simulation, as described in section 2.4. To construct the Jacobian matrix, we explicitly perturbed each individual element of the state vector in independent GEOS-Chem simulations and calculate the sensitivity of $XCH_4$ to that perturbation. We did not use any tagged simulation in this work. This is described Section 2.6.

Line 255: Comparison with NOAA is an important first step to determine whether your posterior solution is consistent with the observed atmospheric growth rate. Unfortunately, Figure 3 and the accompanying text is not sufficiently clear for this reviewer to make that judgement. For the lower right panel of Figure 3 it would be better to use a smaller y axis range given the small differences. At least then a reader can eyeball the comparison. I recommend "better fit" is substituted with some quantitative statistics, e.g. bias, correlations, etc.

We now add the panel with smaller y axis in the Figure S6. We still keep the original panel in the main text (which has the same y axis as the left panel), as it is easy for a reader to compare prior and posterior simulations. We now report root-mean-square errors to quantitatively describe the fit to observations.

Comment on arrangement of paper: showing your ability to independently estimate wetland and non-wetland emissions on the spatial scale you are using (Figure 14) would be useful up front before the more detailed discussions begin. See my comments below.

We now add more discussion on the posterior error correlation before we start presenting inversion results in Section 3 (previously Section 2.7). We have moved previous Figure 14 (Now Figure 5) and relevant discussion to this section. As mentioned in above responses, we also add a new figure (Figure 6) with more detailed discussion on error correlations between wetland and anthropogenic emissions at a regional scale.

A broad comment relevant to more than one figure, e.g. Figure 3: the period 2010 to 2018 covers a wide range of climate variations (e.g. both phases of ENSO) so that taking the mean or differences over this period hides a lot of useful information about how methane has changed.

The bottom panels of Figure 3 already show the monthly time series for the period. The posterior errors (Fig. 3d) are stable throughout the period with reduced seasonal error compared to the prior results. In particular, they show no coherence with ENSO phases. We now modify the relevant text to improve clarity. We also add Fig. S3 in the Supplementary Material, which shows the time series of errors compared to independent ground-based observations. Fig. S3 also shows that the posterior errors have no coherence with ENSO phases.

Line 289: averaging kernel sensitivities? The matrix of averaging kernels already describes sensitivity. Do the authors mean the sensitivities described by this matrix?

We introduced averaging kernel sensitivities (in Section 4.1) as diagonal terms of the averaging kernel matrix corresponding to a specific state vector element, which represent the sensitivity of a state vector element to its true value. The spatial/temporal distribution of this quantity visualizes locations where/when the inversion has strong/weak observational constraints. To better introduce this concept, we now formally define averaging kernel sensitivities in Method section (Section 2.6). We also add more explanations to the captions of relevant figures.

Line 292: from what I understand the state vector has a length of 1000s but the DOFs is limited to 179. What implications does that have for being able to resolve the state vector?

We now add more relevant discussion following the statement. Basically, this indicates that the inversion cannot fully resolve the 1009 state vector element. The constraints from the observations vary spatially, as shown in the figure of averaging kernel sensitivities (Fig. 7 right) mentioned above.

Line 291: "By applying the posterior/prior correction factors to the prior distribution of each anthropogenic emission sector, we obtain improved estimates for anthropogenic emissions for that sector." This statement does not make sense. Improved how? Better fit to measurements? Smaller uncertainties? Applying correction factors does not improve estimates for anthropogenic emissions. Which sector?

The sentence is removed.

Comments on reporting results: some posterior estimates are accompanied by their uncertainties and some are not. The authors need to be consistent. They should certainly report them on Figure 6.

We now report throughout the text the errors derived from posterior error covariance matrix. The error bar is not plotted in Figure 6 and Figure 7 (left panel) because they are often too small compared to the absolute magnitude. We now note this in the figure captions.

Line 315: I am really interested to hear more about how their results point towards an underestimate in livestock emissions. This whole paragraph contains so much hand waving I thought I was at a pre-pandemic music festival. I urge the authors to justify their result in light of it being inconsistent with Maasakkers et al and other published studies that attribute most of these changes to wetlands. I acknowledge they take a second swing at this point on page 15 but I was unsure which they used for their prior (for Figure 8) and more importantly how confidently they could separate wetland and livestock emissions.

Both paragraphs are rewritten to better present the results.

We first would like to clarify that the first paragraph is for mean anthropogenic emissions, and the second paragraph is for emission trends. So, they are separate discussions and are placed in different sections of the paper. We now make changes in both paragraphs to increase clarity.

Our results are consistent with Maasakkers et al. in that both inversions found prior emissions underestimate over these regions. We differ, however, in the attribution. Maasakkers et al. mainly attributed the underestimation to wetland, based on fractions of each sector in prior information. Maasakkers et al. did not exploit the significant seasonal and spatial structure of wetland emissions, which our approach makes use of to separate wetland and non-wetland emissions (with different spatial and temporal resolution). We now add discussion about the error correlations between anthropogenic and wetland emissions (Figure 6). We also present the results of the sensitivity inversion mentioned above, which shows the results presented are robust against the perturbation.

Fig. 10 (previously Fig. 8) discusses the regional emission trends attributed to the livestock sector. We use trends in these inventories to support that increasing livestock emissions over these regions are also plausible from the bottom-up perspective. They are not prior information in our inversion, as we assumed zero prior emission trends in the inversion (described in Section 2.3 and repeated also in the figure's caption). We have modified the text for Fig. 8 to remove any ambiguity.

Line 358: why are the constraints strongest over India and China?
The averaging kernel sensitivities are a function of multiple factors including

observation number, transport, and emissions, and are often large over regions with large emissions. This constraint stems mainly from the fact that these strong emissions, if exist, can create large methane gradients, that are easy to detect by satellite observations. We now briefly explain it in the text.

Line 539: That's a very long lifetime for methane against OH oxidation. This reviewer is left wondering whether the authors' inverse problem remains ill-posed given their state vector. The joint PDF clearly shows strong correlations between different parts of the (global mean) state vector. Given the authors reporting of regional methane missions, it would be useful to show the regional joint PDFs for which I suspect the correlations between anthropogenic and wetland emissions can be much higher. Their conclusion on page 24 doesn't fill me with confidence.

The lifetime we derived is within but at the high end of that derived from the methylchloroform proxy. We now also add text here to remind the readers of the strong error aliasing between global OH and global wetland emissions in this inversion (Figure 5).

As mentioned above, we add Fig. 6 which show that the error correlations between anthropogenic and wetland emissions are moderate at regional level (r: averages -0.3; range -0.1 to -0.5).

Reviewer #2

This article analyses the role of wetland and livestock on the global CH4 growth rates in the period 2010-2018. The authors have also carefully assesed the role of OH on the CH4 growth rate. The article is generally well written, and has built upon their earlier analysis. I have doubt on the simplicity of the emission optimisation for only two major sectors, while CH4 has so many other natural and anthropogenic emission sectors those vary very differently at interannual and longer timescales. Detailed comments below. The manuscript may be considered for publication after addressing some of these comments.

Specific comments: lines 38-39 : need references?
A reference is added.

lines 76ff: I find later that the CTM configuration and other details are discussed later. May be remove this paragraph

We remove the paragraph.

lines 90ff: I think this is not an ideal choice because the values of $CH_4$ depends on assumed $CO_2$ concentrations. We know that our understanding of $CO_2$ fluxes over many data void regions are poor. How do you estimate regional bias, say over South Asia, Southeast Asia, Amazon/Brazil, Africa etc.

The $CO_2$ proxy retrieval has been systematically evaluated against ground-based measurements with its uncertainty characterized (Parker et al., 2020) and has been extensively used in other methane inversion studies. In particular, validation has also been performed for the uncertainties induced by the model XCO2 (Parker et al., 2015). An evaluation was also performed over Amazon against aircraft profile observations (Webb et al., 2016). We now add information about these validation studies in the manuscript.

line: that's the lower bound, what about the oceans?

We do not know what this comment is referred to.

line 137: Did you apply any scaling to termite emissions?

We did not apply any scaling to Fung et al. (1991) in the prior emissions. We do optimize termite emissions together with other non-wetland emissions. We now report the total termite emissions in the text in case readers are interested in the magnitude.

lines 170ff: 1-year spin-up is too short by any standards for long-lived species. What about the stabilisation of vertical gradients? You cannot get that right by the unbiasing method you describe. For future studies please make a spin-up for 10 years or so. Its worth the computing cost, given the large amount of follow work and analysis is done for any publication.

The previous description was inaccurate. The 1-year spin-up simulation was initialized from a $CH_4$ field generated from a long-term simulation (Turner et al., 2015), in which stabilization of global-scale methane gradients is already achieved. The additional 1-year simulation using 2009 meteorology was to establish reasonable synoptic-scale

gradient on the starting date (Jan 1 2010). We now modify our description to avoid this confusion.

Figure 2 and associated text: Does these correction factors extend till the poles? The problem is that the stratosphere is dynamic as well. Are such seasonal bias correction factor good for getting the profiles right.

We fully agree with you that the stratosphere is dynamic. We do find that the correction factors are better predicted by equivalent latitude rather than ordinary latitude. Equivalent latitude is a function of potential vorticity, which partly reflects the dynamics of stratosphere, in particular polar vortex. We now emphasize this point in the text. It is not necessary to extend the correction to the poles, because we do not have any GOSAT data over very high latitudes to begin with.

My worry here is also that if the stratosphere is not spun-up well these bias correction factors will not be time invariant. So the trends in anthropogenic or natural emissions you derive later may not be free of these correction factors.

See above responses on initial conditions for "not well spun-up stratosphere". In addition, we now also compute the correction factors by individual years, and find no significant shift in the correction factors (Figure S2).

Figure 5 and associated text: Very tough to accept the results for the a posteriori patterns, e.g., there is a strip of increased emissions along the Himalayan region! Is this arising from not properly accounting for the orography in the coarse resolution model? Or this may be an artifact of the proxy retrievals by miscalulating $CO_2$ over northern part of the Indian subcontinent.

The absolute changes along the Himalayan region are very small (because prior emissions there are very small). We now add Figure S7, which shows the absolute changes between prior and posterior estimates to help readers interpret the results. We still retain the posterior/prior ratio plot, as the ratios are useful for interpreting results for regions with large emissions.

Figure 6 and associated text: I like this analysis but it is not clear at all, if the results are independent of the priors! For example, if you if you started with the priors from the UNFCCC will you get the same a posteriori emissions? The question is also same for

the trends in derived emissions. Some sensitivity tests are need for clarifying robustness of the a posteriori emissions. See for example Patra et al., JMSJ, 2016. That paper is also relevant for other discussion in the paper where you discuss trends of emissions over China, and from Animals etc.

We now report the reduced averaging kernel sensitivities in the figure, which characterizes the dependence of the results on observations vs. prior. We now also cite Patra et al., 2016 and include it for discussion where relevant.

lines 340-344: How confidently we can talk about the oil and gas emission trends - given that the cited references are so small scale compared to the gridcell you optimize here. For example, can you gather the cited references by model gridcells and check the validity of the inversion and vice versa?

Though in coarse resolution, the upward correction patterns over Mid- and South U.S. in Fig. 7 correspond to major U.S. oil and gas production regions, where underestimation of emissions is reported by work of these cited references repeatedly. We now add a discussion on Maasakkers et al. (2020) who performed a high-resolution inverse analysis over the U.S. using the 2010-2015 GOSAT data, and allocated the correction more specifically to these oil/gas regions.

lines 366-377: I am a bit confused why/how all the anthropogenic emissions are linked with the livestock population? Is this because your a priori emissions only accounts for the livestocks? For example how is the trends in waste management in these developing nations?

We now modify the text to avoid any confusion that the attribution is made based on livestock population. We attribute these trends to emission sectors based on their fractions in prior information over a grid cell. We then support this attribution with bottom-up information (both livestock population data and bottom-up livestock inventories).

Page 18-20: I do not know but I have a feeling that the inversion is set in such a way that all the regions are having somewhat similar increase in emissions, either from the animals or from wetlands. The authors are in the best position to judge and find the reasons behind such outcomes from their inversion system.

Fig. 9 shows the spatial patterns of anthropogenic emission trends derived from our

inversion. Fig. 12 shows the trends of wetland emissions in 14 subcontinental regions. Both show that varied magnitudes of trends are inferred for different regions by our inversion.

If you can explain the global CH4 growth rates using emissions from only two emission sectors then we have an oversimplifications of CH4 sources. I cannot prove anything in favour or against the proposed mechanisms here but many of the hypothesis are based on apparently single line of evidence, i.e., the GOSAT proxy XCH4 retrievals. Aren't Maasakkers et al., Lunt et al., Pandey et al., Parker et al. using the same XCH4 data sources?

We did not intend to state that wetland and livestock are the only two sectors contributing to accelerating methane growth. We now make several changes in the text to avoid this misimpression. In particular, when discussing wetland contributions, we are now more explicit about the years (2016-2018) that we infer large wetland emissions, rather than presenting them as a trend (which is misleading because they are just large inter-annual variability occurring in the latter part of the study period). We also discussed contributions from other sectors (Fig. 13 and relevant text).

Indeed, evidence are mostly from GOSAT XCH4 data for now, because of the lack of other measurements in these remote regions. Our findings suggest that more attention should be paid to these regions.

In addition to the trends analysis, it would have been useful to discuss how and why are the large interannual variability in some regions as shown in Fig. 10. Are there irregular data gaps or a particular climate anomaly affecting CH4 emissions regionally?

Thanks for the suggestions. We have a paper in preparation that uses the satellite-constrained wetland emissions to examine the drivers of their variability in different regions.

line 470: The NH/SH OH ratio of 1.02 is an interesting result from this exercise. Could you please comment here how are the NH/SH ratio of CH4 emission changed from a priori to a posteriori cases? I suspect the derived NH/SH OH ratio depends on how well freely the inversion system is allowed to adjust emissions vs that for OH.

Our setup is based on the reasoning that emissions can be informed by

regional/subcontinental gradients of CH$_4$ while OH mainly is informed by global/hemispheric gradients of CH$_4$. With this setting, both emissions and OH have some freedom to adjust. The figure below shows the NH/SH ratio of emissions in prior and posterior estimates, which demonstrates that hemispheric emission ratios are indeed being adjusted by the inversion (though emissions are resolved at much finer resolution). We decide not to include the figure in the manuscript to better focus on the OH ratio results.

[Figure]

page 23: How can we get assured that the OH and CH4 emissions can be optimised in one inversion system. Is there a dipole effect? For example, if you do more or less number of iterations, will the global total emission and global mean OH will be different? I see that this issue is addressed later using Fig. 14, but still not convinced that the inversion system is separating the wetland vs livestock emissions well.

We solve the inversion analytically. So, we do not iterate to find the solution. In our previous work (Zhang et al., 2018), we tested separate optimization of OH and CH4 emissions in an Observation System Simulation Experiment, which demonstrated some ability to separate changes in OH and CH4 emissions in an inversion similar to the setting here. Zhang et al. (2018) identified that the major source of uncertainty to the method is the prescribed distribution of spatial-temporal OH fields (about 3% systematic uncertainty). We now add text to remind readers that with current observations we have only limited ability to separate OH and emissions.

As described in the above responses, we now add more discussion to support the separability of wetland and anthropogenic emissions on regional scales. Briefly, we add

a new figure (Figure 6) to show the posterior error correlations between regional wetland and anthropogenic emissions. We also perform a sensitivity inversion to test the impact of unresolved subregional wetland distribution on posterior wetland and anthropogenic estimates. In this sensitivity simulation, prior wetland distribution in Sudd and the Congo Basin of Africa (which are near livestock hotspots) is perturbed (Figure S1). Results are shown in Figure S4, Figure 8 (anthropogenic emissions), and Figure S5 (wetland emissions).

Figure 13: You attributed emissions from livestock increase to the animal population, but what appears here is that the increase in emissions from tropical and extratropical wetland are the greatest. Could you propose a mechanistic viewpoint?

We have briefly included some discussion on the driver based on literature. As mentioned above, we will explore from this perspective in a paper in preparation.